**Investigation**

# Testing times: disentangling admixture histories in recent and complex demographies using ancient DNA

Matthew P. Williams [1,*] Pavel Flegontov [2,3] Robert Maier,[3] Christian D. Huber [1,*]

[1]Department of Biology, Pennsylvania State University, University Park, PA 16802, USA
[2]Department of Biology and Ecology, University of Ostrava, Ostrava 701 03, Czechia
[3]Department of Human Evolutionary Biology, Harvard University, Cambridge, MA 02138, USA

*Corresponding author: Department of Biology, Pennsylvania State University, University Park, PA 16802, USA. Email: mkw5910@psu.edu;
*Corresponding author: Department of Biology, Pennsylvania State University, University Park, PA 16802, USA. Email: cdh5313@psu.edu

Our knowledge of human evolutionary history has been greatly advanced by paleogenomics. Since the 2020s, the study of ancient DNA has increasingly focused on reconstructing the recent past. However, the accuracy of paleogenomic methods in resolving questions of historical and archaeological importance amidst the increased demographic complexity and decreased genetic differentiation remains an open question. We evaluated the performance and behavior of two commonly used methods, qpAdm and the $f_3$-statistic, on admixture inference under a diversity of demographic models and data conditions. We performed two complementary simulation approaches—firstly exploring a wide demographic parameter space under four simple demographic models of varying complexities and configurations using branch-length data from two chromosomes—and secondly, we analyzed a model of Eurasian history composed of 59 populations using whole-genome data modified with ancient DNA conditions such as SNP ascertainment, data missingness, and pseudohaploidization. We observe that population differentiation is the primary factor driving qpAdm performance. Notably, while complex gene flow histories influence which models are classified as plausible, they do not reduce overall performance. Under conditions reflective of the historical period, qpAdm most frequently identifies the true model as plausible among a small candidate set of closely related populations. To increase the utility for resolving fine-scaled hypotheses, we provide a heuristic for further distinguishing between candidate models that incorporates qpAdm model $P$-values and $f_3$-statistics. Finally, we demonstrate a significant performance increase for qpAdm using whole-genome branch-length $f_2$-statistics, highlighting the potential for improved demographic inference that could be achieved with future advancements in $f$-statistic estimations.

Keywords: ancient DNA; aDNA; archaeogenetics; paleogenomics; qpAdm; $f$-statistics; admixture

## Introduction

Beginning over a decade ago, the genome sequencing and analysis of ancient specimens, so-called ancient DNA (aDNA), spawned the field of paleogenomics and have provided novel insights into our understanding of population demographic history for a diversity of organisms and contexts (Brunson and Reich 2019; Schepper et al. 2019; Spyrou et al. 2019; Arning and Wilson 2020; Mitchell and Rawlence 2021; Wibowo et al. 2021). No species has gained deeper insights from the aDNA revolution than humans, as it has significantly unraveled our complex evolutionary and migratory histories (Fu et al. 2016; Haber et al. 2016; Slatkin and Racimo 2016; Llamas et al. 2017; Williams and Teixeira 2020; Liu et al. 2021; Ávila-Arcos et al. 2023). Much of the research in human paleogenomics during the early 2010s was focused on reconstructing human prehistory (dating back more than 5,000 YBP; Fig. 1a). It was during these years that many of the statistical methods and software that have since become the foundation of aDNA studies were developed and have been pivotal in defining our understanding of human prehistory. These methods range from model-free exploratory approaches, such as principal component analysis and smartpca implementation (Patterson et al. 2006; Reich et al. 2008; McVean 2009) and the ADMIXTURE software (Alexander et al. 2009), to statistical tests of admixture such as $f_3$- and $f_4$-statistics (Reich et al. 2009; Patterson et al. 2012) and the related D-statistics (Green et al. 2010; Durand et al. 2011), which leverage deviations from expected allele sharing patterns to reject simple trees and suggest more complex relationships. In addition, various downstream software has been developed to elucidate more complex relationships among numerous groups, with many utilizing $f$-statistics. Examples include qpAdm, which models a target population as a mixture of several proxy ancestry sources (Haak et al. 2015; Harney et al. 2021); qpWave, analyzing the number of gene flow events between population sets (Reich et al. 2012); and qpGraph, MixMapper, TreeMix, AdmixtureBayes, and findGraphs, all creating representations of admixture histories as directed acyclic graphs (Patterson et al. 2012; Pickrell and Pritchard 2012; Lipson et al. 2013, 2014; Maier et al. 2023; Nielsen et al. 2023). To a large degree, the reliance on these methods has been because of their use of allele frequencies which is suitable for pseudohaploid aDNA whereby calling diploid genotypes is often infeasible due to its highly degraded characteristics.

Since the 2020s, there has been a shift in aDNA research to studying the more recent past (Fig. 1a). As a result, aDNA is increasingly used to address questions of archaeological and historical relevance. This research field was named archaeogenetics by

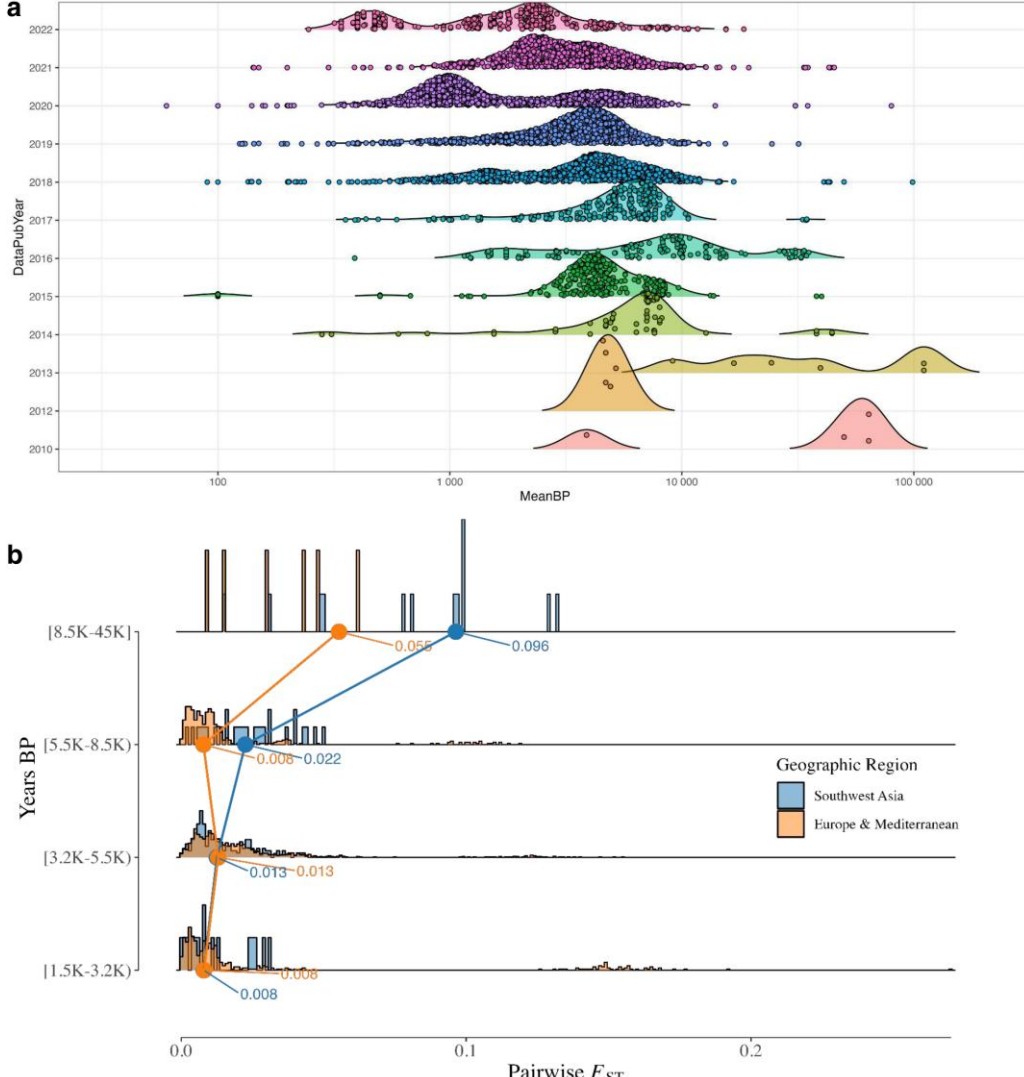

**Fig. 1.** Dates of published aDNA samples. a) A per-publication-year transect of the density of the (log10) age of published ancient genomes. The publication dates and number of samples were taken from the Allen Ancient DNA Resource (AADR) v.52.2. b) A temporal transect of population differentiation levels in Southwest Asia, Europe, and the Mediterranean inspired by Lazaridis *et al.* (2016). The average dates for each sample in YBP were taken from the AADR v.52.2. For the plot in b), they were grouped into four epochs, with 3,200 YBP approximating the start of the Iron Age, 5,500 YBP approximating the start of the Bronze Age, 8,500 YBP approximating the start of the Neolithic period, and older years representing the Paleolithic period. For each epoch, the median pairwise $F_{ST}$ values are displayed for the two regions. The $F_{ST}$ values were calculated using the Eigensoft v8.0.0 smartpca software.

British archaeologist Colin Renfrew (Renfrew and Boyle 2000). The historical period, particularly in Southwest Asia, is broadly demarcated to begin somewhere around the early-mid-3rd millennium BCE (Bartash 2020) and is characterized by the invention of writing and intermittent periods of intensified inter-regional trade, diplomacy, and human mobility (Kristiansen 2016). From this body of research, hypotheses about admixture between ancient settlements amenable to aDNA can involve groups separated by very short periods of time and thought to have descended from a complex web of migration and population structure (Haak *et al.* 2015; Lazaridis *et al.* 2016, 2017, 2022a, 2022b; Haber *et al.* 2017, 2020; de Barros Damgaard *et al.* 2018; Harney *et al.* 2018; Antonio *et al.* 2019; Narasimhan *et al.* 2019; Wang *et al.* 2019; Agranat-Tamir *et al.* 2020; Fernandes *et al.* 2020; Skourtanioti *et al.* 2020, 2023; Clemente *et al.* 2021; Koptekin *et al.* 2023; Moots *et al.* 2023; Schmid and Schiffels 2023). These can range from questions regarding the degree of population continuity between periods of cultural change or

settlement hiatus in the archaeological record to determining if cultural links between regions are indicative of inter-regional migration. Additionally, they may include assessing if historical records of mass migrations and forced relocations result in observable signals of increased inter-regional gene flow. A common thread underlying these questions is, for a population of interest, to what extent can aDNA accurately reconstruct their genetic history and, importantly, reject false models of ancestry composed of closely related candidate populations? Moreover, what limits and possible biases emerge with the increase in demographic complexity among candidate source populations, a reduction in the number of generations separating aDNA samples and their ancestral admixture events, and an overall decrease in genetic differentiation indicative of the historical period? While the theoretical behavior of *f*- and D-statistics has been extensively tested (Patterson *et al.* 2012; Martin *et al.* 2015; Peter 2016, 2022; Harris and DeGiorgio 2017; Zheng and Janke 2018; Soraggi and Wiuf 2019; Tricou *et al.* 2022) and the performance of the

commonly used software qpAdm thoroughly assessed under simple demographic models with both pulse-like and continuous migration (Ning *et al.* 2020; Harney *et al.* 2021), their behavior under varying degrees of population differentiation and complex demographic history expected of populations within the historical period remains underexplored.

In this study, we conducted a simulation-based evaluation of two widely used methods for reconstructing admixture histories—the "admixture" $f_3$-statistic and the qpAdm software (Fig. 2). Our goal was to understand their effectiveness and limitations, particularly in complex scenarios that arise during the reconstruction of historical population dynamics. We started by simulating two chromosomes of combined length ~491 Mb under four simplistic and qualitatively different admixture graphs, aiming to explore a broad range of model parameters leading to widely varying degrees of genetic differentiation. Subsequently, we expanded our evaluation to include a complex demography representative of a model of Eurasian human history emerging from a series of recent publications, which comprised 59 populations and 41 pulse admixture events. We simulated 50 whole-genome ($L$ ~2,875 Mb) replicates and processed the simulated data to mimic typical aDNA conditions, including a Human Origins–like SNP ascertainment scheme, empirical data missingness distributions, and pseudohaploidization.

Importantly for the study of the historical period, our findings illustrate that as $F_{ST}$ levels reach those observed in Bronze and Iron Age European, Mediterranean, and Southwest Asian groups, qpAdm converges on a small subset of plausible models for an admixed target group consisting of the true sources and closely related populations. However, under these divergence levels and conditions typical of aDNA, we observe that qpAdm has limited ability to definitively answer fine-scaled questions relevant for archaeologists and historians due to a lack of power to reject all nonoptimal ancestry sources minimally differentiated from the true ones. Moreover, for historical populations with complex gene flow histories, we show that while admixture to source populations generally improves the performance of qpAdm, the phylogenetic origin of this admixture in ancestral source groups differentially impacts qpAdm accuracy and performance. We show that the number of generations postadmixture has no impact on qpAdm performance or accuracy of admixture proportion ("admixture weight") estimates. However, we observe when selecting suboptimal ancestry sources that the admixture weights are biased in favor of the population that is most similar to the true source. We assessed several model plausibility criteria commonly used in the aDNA literature and show that each criterion impacts the performance and accuracy of qpAdm differently under various demographic conditions. Additionally, we highlight challenges that users should be aware of when applying additional plausibility criteria for qpAdm models, such as negative admixture $f_3$-statistics or the rejection of all simpler qpAdm models, as they can lead to an increase in type II errors. Finally, we offer an interpretative heuristic guide that can enhance the power to distinguish between multiple plausible qpAdm models, thereby increasing the utility of archaeogenetic inference in reconstructing historical demographic history.

## Materials and methods
### Simple demographic models: parameter configurations

To obtain a baseline understanding of how specific demographic models and parameters impact downstream population genetic

inference with qpAdm and the $f_3$-statistic, we formed simple bifurcating trees with varying scales of population divergence and augmented them with one to three gene flows in qualitatively different configurations. We note that all our simple demographic models described below do not violate the topological assumptions of qpAdm outlined in Harney *et al.* (2021) (Fig. 3a–d) and all samples were taken at the "present." For the simplest bifurcating demographic model with one admixture event (hereafter Model 1; Fig. 3a), we randomly sampled values of five split-time parameters ($T_1$, $T_2$, $T_3$, $T_4$, and $T_{admix}$) from uniform distributions generated by the following framework:

- The oldest variable split-time ($T_1$) was selected first from a window between four generations in the past and the fixed $T_0$ split-time (6,896 generations).
- The $T_2$ split-time parameter was sampled between three generations in the past and the sampled $T_1$ split-time.
- We selected the $T_3$ and $T_4$ split-time parameters from a window between two generations in the past and the $T_2$ split-time parameter.
- The $T_{admix}$ (admixture date) parameter was selected from a window between a single generation in the past and the minimum of the $T_3$ and $T_4$ split-time parameters.
- We randomly sampled the admixture weight parameter ($\alpha$), which forms the target population as a mixture of the Source 1 (proportion $\alpha$) and Source 2 (proportion $1 - \alpha$) populations, from a uniform distribution between 0 and 1 (the distributions of simulated parameter values and scatterplot matrices of simulation parameter correlations can be found in Supplementary Fig. 1a and b).

To assess the impact on admixture inference of more complex admixture history in one of the proxy ancestry sources, we configured three additional demographic models, each building upon the structure of Model 1 as follows:

- Model 2 includes a gene flow from an out-group (R3 branch) into the source (S1).
- Model 3 includes admixture into the source (S1) from an internal branch ancestral to both the S2 and R2 populations (iS2R2).
- Model 4 combines the admixture events from Models 2 and 3, with no constraint on their order.

### Simple demographic models: data generation and analysis

For each of the four simple demographic models (Fig. 3a–d), we used msprime v.1.2.0 (Kelleher *et al.* 2016; Baumdicker *et al.* 2022) to simulate 5,000 iterations of succinct tree sequences without mutations with each iteration sampling demographic parameters from the schema outlined above. To accurately capture the impacts of long-range linkage disequilibrium driven by recent admixture, we used a two-phase process whereby for the first 100 generations into the past, we simulated under the discrete-time Wright–Fisher model (DTWF; Nelson *et al.* 2020) and then under the Standard (Hudson) coalescent model until the most recent common ancestor (MRCA). We used sequence lengths and recombination rates approximating human chromosomes 1 ($L = \sim2.49 \times 10^8$ bp and $r = \sim1.15 \times 10^{-8}$ per bp per generation) and 2 ($L = \sim2.42 \times 10^8$ and $r = 1.10 \times 10^{-8}$; Adrion *et al.* 2020; Lauterbur *et al.* 2023) and separated each chromosome with a log(2) recombination rate following guidelines in the msprime manual (https://tskit. dev/msprime/docs/stable/ancestry.html#multiple-chromosomes). For each demographic model and for all populations, we fixed an

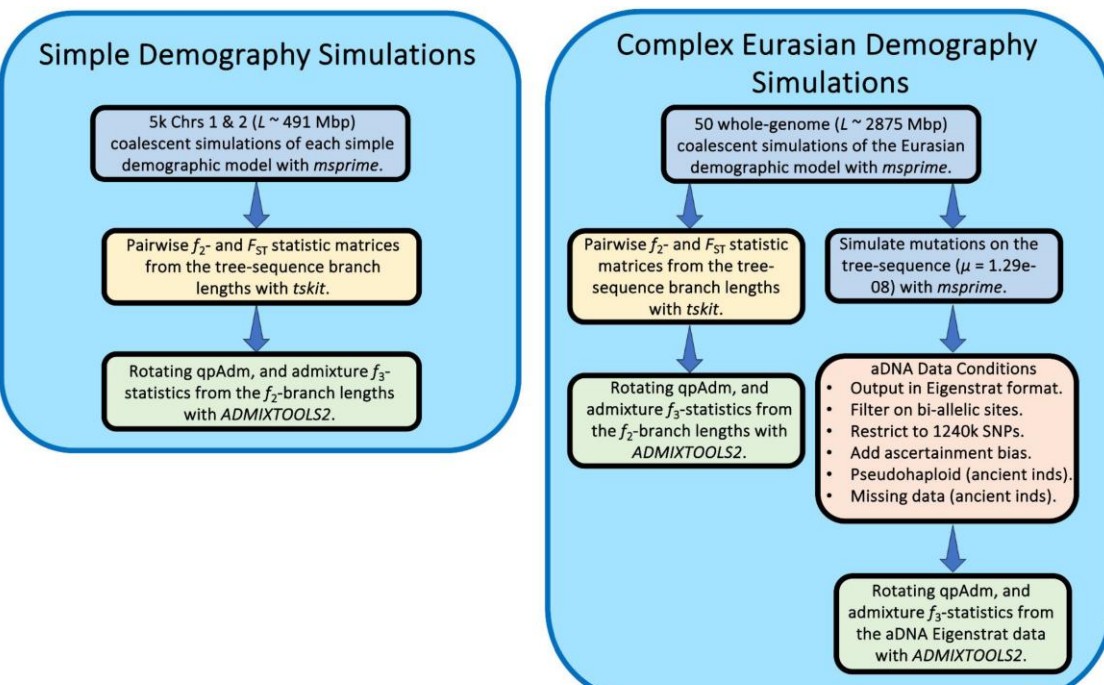

**Fig. 2.** Simulation and analysis workflow in our study.

effective size ($N_e$) of 10,000 and a sample size of 20 diploid individuals taken at the leaves.

We generated $f_2$ - and $F_{ST}$-statistic matrices directly from the tree sequences through tskit v.0.5.2 with parameters "Mode = branch" and "span_normalise = True," using 5 Mb windows. The resulting $f_2$-statistics matrices were used for qpAdm analyses with parameters "full_results = TRUE" and "fudge_twice = TRUE" and for calculating admixture $f_3$-statistics in the ADMIXTOOLS2 software (Maier *et al.* 2023). For the qpAdm rotating protocol following Harney *et al.* (2021), we included S1, S2, R1, R2, R3, and R4 as alternative sources and "out-groups" ("right" populations), resulting in six single-source and 15 two-source models. We computed admixture $f_3$-statistics on pairwise combinations of the S1, S2, R1, R2, R3, and R4 populations, resulting in an $f_3$-statistic test for each of the 15 two-source qpAdm models.

Throughout our analysis of the simple demographic models, we refer to the pairing of the S1 + S2 source populations as the "true" model representing the ancestry of the target population, and we refer to all other population combinations as "false" models. In evaluating the qpAdm results, unless otherwise stated, we consider plausible models to have a $P$-value $\geq 0.05$ and admixture weights between 0 and 1 ([0,1]). In addition, we configured a summary metric, "qpAdm test performance" (QTP), that conveys the precision of rotating qpAdm analyses per simulation iteration taking into account all single- and two-source qpAdm models (Fig. 3e–h). The range of QTP is between "+1" and "−1" where the most optimal outcome, "+1," corresponds to the condition where all false models are rejected (single- and two-source), and only the true model is plausible. The least optimal outcome, "−1," occurs when the true model is rejected, and all false models are considered plausible. As such, all rotating qpAdm analyses that reject the true model result in a negative QTP, and analyses that include the true model among the plausible qpAdm candidates have positive QTP values. The outcomes where all models are rejected, or all models are plausible, are scored as "0." Values between "+1" and "0"

occur when both the true and false modes are plausible in the same simulation, with each additional plausible false model (single- and two-source) decreasing the QTP value. Likewise, values between "0" and "−1" occur when the true model is rejected, and some (but not all) false models are plausible. We also evaluated the binary QTP outcome (Fig. 3i–l), whereby qpAdm either performs most optimally (i.e. rejects all false models and estimates the true model as plausible) or does not (i.e. at least one wrong model is considered plausible, or the true model is rejected).

## Complex demographic model: parameter configurations

We expanded our evaluation of admixture inference from simple topologies to a demographic model and data distribution that more closely reflects the real-world complexities of both Eurasian human history and aDNA conditions. We framed this by simulating an archaeogenetic hypothesis on the origin of migrants to the Southern Levant at the beginning of the Iron Age (the so-called Sea Peoples migration). While our demographic model and parameters are informed by the aDNA and population genetic literature, we stress that it is not designed to represent true human history, nor a proposal of the likely events associated with the purported Sea Peoples migration. Rather, its function is solely to capture some of the complexities surrounding the dynamics connecting populations in the historical period such as low divergence between candidate source populations, complexity of ancestral population relationships, and sampling recently after the admixture event. As such, it provides us a framework from which we can evaluate the behavior and limitations of admixture inference under complex demographic models using aDNA.

In total, we model 59 populations and 41 pulse admixture events (see Fig. 4a for a simplified topology) which are all described and referenced in Supplementary File 1 (Lazaridis *et al.* 2016; Damgaard *et al.* 2018; Lazaridis *et al.* 2018; Fregel *et al.* 2018; Harney *et al.* 2018; Antonio *et al.* 2019; Feldman *et al.* 2019; Narasimhan *et al.* 2019;

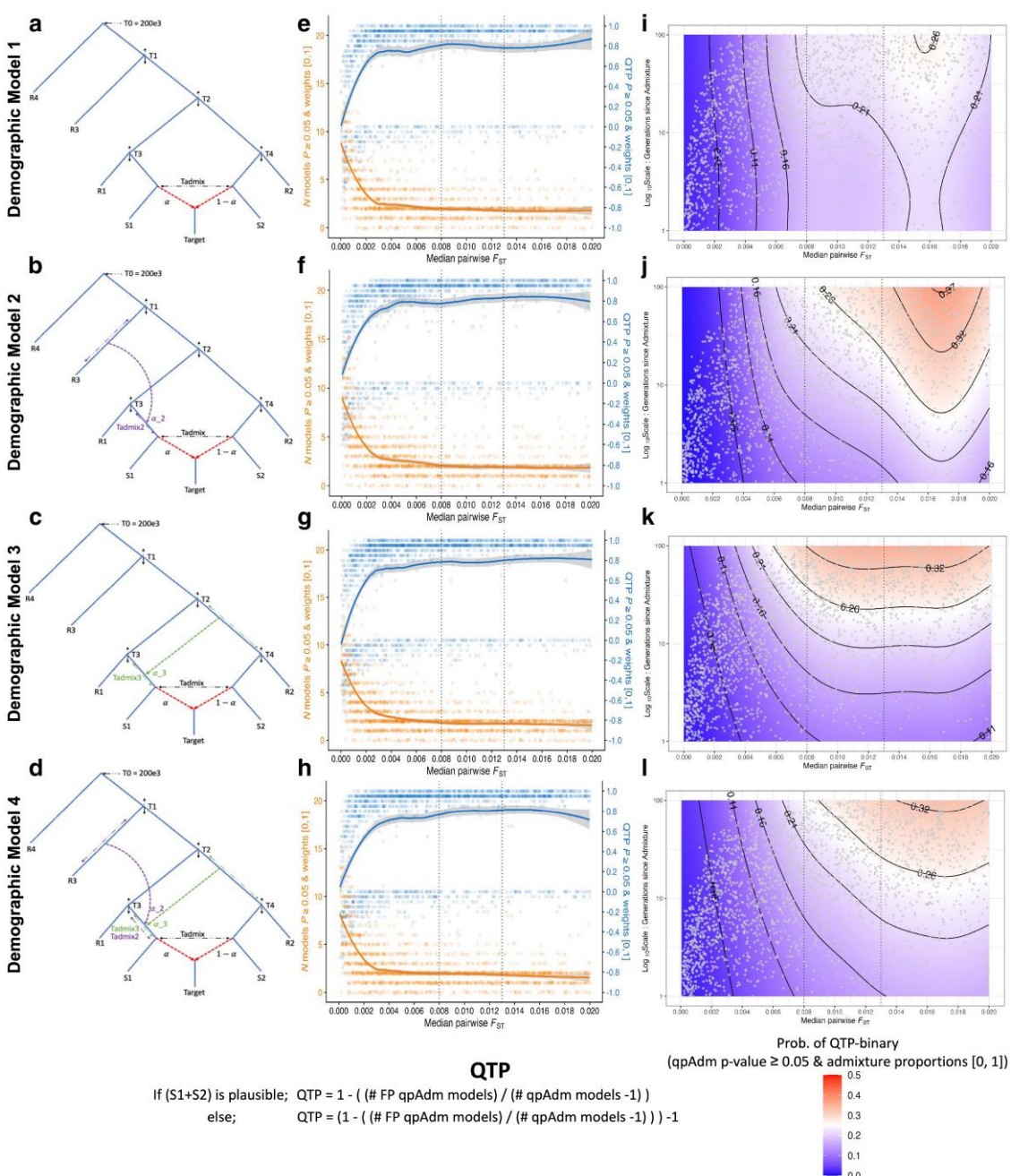

**Fig. 3.** Simple demographic models and QTP. a–d) Topological structures of the four simple demographic models. e–h) QTP and number of plausible qpAdm models across the historical range ([0, 0.02]) of median pairwise $F_{ST}$ values calculated on the S1, S2, R1, R2, and R3 populations and ≤100 generations since the admixture event forming the target population. For each simulation iteration, we represent the counts of the number of plausible single- and two-source qpAdm models (21 is the maximum possible) with orange dots and the locally estimated scatterplot smoothing (loess) computed in R and shown with the orange line. We show the QTP value for each simulation iteration with blue dots and the loess smoothing with the blue line. i–l) Logistic generalized additive model probability for the QTP-binary response variable with admixture date ($T_{admix}$) and median pairwise $F_{ST}$ as predictor variables (parameters restricted to the historical range described above). The gray dots are unique combinations of simulation parameters placed in the space of predictor variables. Vertical dotted lines in plots e–l show the median pairwise $F_{ST}$ values at the approximate Iron Age (0.008) and Bronze Age (0.013) periods.

Wang *et al.* 2019; Bergström *et al.* 2020; Fernandes *et al.* 2020; Kamm *et al.* 2020; Skourtanioti *et al.* 2020; Clemente *et al.* 2021; Hollfelder *et al.* 2021; Marchi *et al.* 2022). We note that our sampling strategy for the simulated aDNA, while reflecting ages and sample sizes of empirical data, does not violate the qpAdm assumptions outlined in Harney *et al.* (2021) and discussed in our companion manuscript (see Discussion and (Yüncü *et al.* 2023). A brief summary of the model scaffold follows. The oldest split in the demography is the

separation of East and Central African ancestral populations at 5,172 generations before present (Hollfelder *et al.* 2021). We model an out-of-Africa population as separating from the East African lineage at 3,303 generations (Kamm *et al.* 2020; Marchi *et al.* 2022) and from the former lineage split East Eurasian, North Eurasian, West European Hunter–Gatherer (WEHG), and ancestral Near Eastern lineages (Kamm *et al.* 2020; Marchi *et al.* 2022). Two meta Near Eastern lineages, eastern and western Near East, split from the

ancestral Near Eastern lineage (Marchi *et al.* 2022). The Levant lineage, from which the target Southern Levant IA population ("sLev_IA1") largely descends, splits from the "western Near East" lineage at 483 generations (Broushaki *et al.* 2016; Lazaridis *et al.* 2016; Marchi *et al.* 2022). We model the formation of a "northwestern Near East" lineage at 446 generations, from which the admixing source population "Aegean Island" ("AegeanIsl_BA") largely descends, as a mix of "western Near East" (0.86) and WEHG (0.14; Marchi *et al.* 2022). The target lineage, sLev_IA1, was modeled as a mixture of its ancestral population (Southern Levant Bronze Age, "sLev_BA") and the AegeanIsl_BA population (admixture fraction = 0.2) at generation 111 before present. We sampled the target population five generations postadmixture (Supplementary File 1). To assess the influence of post-admixture drift on admixture inference, we modeled successive step-wise splits from the target lineage and sampled them 10, 25, 50, 80, and 100 generations post the original admixture event. From our simulated lineages, we sampled data representing the Mbuti present-day population and 20 ancient Eurasian and African populations that reflect empirical ancient groups present in many Southwest Asian aDNA analyses (Fig. 4b). For all populations, we sampled 10 individuals, and in all downstream analyses, we defined the pairing of sLev_BA + AegeanIsl_BA as the true model and all others as false models. As above, we consider plausible models to have a *P*-value ≥ 0.05 and admixture weights between 0 and 1 ([0,1]) unless otherwise specified.

## Complex demographic model: data generation and analysis

We configured the Eurasian demographic model described above using the Demes graph format (Gower *et al.* 2022) and converted it to an msprime demography object through the demography.from_demes() function. We simulated 50 whole-genome (*L* ~2,875 Mb) replicates using sequence lengths and recombination rates of chromosomes 1–22 following the HomSap ID from the stdpopsim library (Adrion *et al.* 2020) and separated each chromosome with a log(2) recombination rate following msprime manual guidelines (Nelson *et al.* 2020, Baumdicker *et al.* 2022). As in the simple demographic model simulations, to capture long-range correlations across the genome due to recent admixture, we simulated the first 25 generations into the past under the DTWF model (Nelson *et al.* 2020). We then simulated under the Standard (Hudson) coalescent model until the sequence MRCA. Our whole-genome simulations required an average of 12.6 ± 2.2 GB of memory and a duration time of 221 ± 10 h per replicate. We applied mutations to the simulated tree sequence at a rate of 1.29*e*−08 (Jónsson *et al.* 2017) using the Jukes–Cantor mutation model (Jukes and Cantor 1969). From the mutated tree sequence, we generated Eigenstrat files through the tskit v.0.5.2 TreeSequence.variants() function which were passed to custom R scripts to generate realistic aDNA conditions such as filtering on biallelic sites, adding ascertainment bias, downsampling to 1,233,013 SNPs (1,240,000 capture), and, for the simulated ancient individuals, generating pseudohaploid data with empirical missing rates (GitHub Repo: https://github.com/archgen/complex_demog_sims.git).

We configured the SNP ascertainment bias scheme replicating the general principles of the Human Origins array (Patterson *et al.* 2012; see also Flegontov *et al.* (2023) for an overview of effects of this type of ascertainment on *f*-statistics and related methods). In the Eurasian demography, we defined separate lineages representing Central European, East Asian, African, and South Asian populations, sampled a single individual from these lineages at

the present, and retained biallelic sites that are heterozygous in at least one of these individuals. We then downsampled the simulated data by randomly sampling 1,233,013 SNP loci. For the simulated ancient samples (Fig. 4b), we randomly assigned one of the two alleles as homozygous at simulated heterozygous positions mimicking what is commonly performed for low- and medium-coverage aDNA (Schuenemann *et al.* 2017). In addition, we added missing data by assigning to each ancient individual an empirical missingness distribution from a randomly selected ancient individual within the Allen Ancient DNA Resource (AADR v.52.2; Mallick *et al.* 2024), which we filtered by removing related and contaminated ancient individuals and restricted to individuals from Southwest Asia (see Supplementary File 2 for the list of empirical aDNA individuals). Among the target populations, this resulted in a range of missingness within each replication (the median standard deviations of the population missingness across the replicates ranged from 0.04 to 0.14), with the average proportion of missingness across the 50 replicates ranging from a minimum of 49% for the sLev_IA3 population to 89% in the sLev_IA1 population (resulting in a range of approximately 130,656–627,948 useful SNPs, respectively; Supplementary Table 2). We observe similar degrees of missingness for the other ancient populations included in the qpAdm rotation analysis (Supplementary Table 2). We generated two additional missing data subsets following the method above, whereby the AADR individuals were filtered to contain only low (SNPs < 100,000), or medium coverage (100,000 < SNPs < 500,000) from samples across Eurasia, resulting in 50 whole-genome replicate simulations with three different degrees of missingness.

From the simulated aDNA, we computed rotating qpAdm analyses with the ADMIXTOOLS2 software (Maier *et al.* 2023) using parameters typical of empirical aDNA workflows such as "allsnps = TRUE" (using all SNPs available for calculating each individual *f*₄-statistic) and 5 Mb windows for calculating standard errors of *f*₄-statistics with the jackknife procedure. We configured the qpAdm analysis protocol in the following way: the most ancient groups are fixed in the right group position and younger candidate populations are rotated between the left and right group positions (Narasimhan *et al.* 2019; Lazaridis *et al.* 2022a). The Mbuti population was fixed in the first position in all qpAdm analyses, along with nine deeply divergent Eurasian and African populations fixed in the right group (Fig. 4b). We then rotated nine simulated Bronze Age and Chalcolithic and two European hunter–gatherer populations (Fig. 4b) between the left and right group positions resulting in a total of 11 single-source models and 66 two-source models.

To evaluate the impact of aDNA conditions on admixture inference under the Eurasian human demography, we generated *f*₂- and $F_{ST}$-statistic matrices directly from the tree sequence without mutations through tskit v.0.5.2 with parameters "Mode = branch" and "span_normalise = True," using 5 Mb windows. The resulting *f*₂-statistics matrices were used to compute rotating qpAdm analyses following the same approach for the aDNA application and admixture *f*₃-statistic in the ADMIXTOOLS2 software (Maier *et al.* 2023).

## Results

### Starting simple: insights into the behaviors of qpAdm and the *f*₃-statistic from simplistic demographic models

Our simple demographic model simulations resulted in genetic diversity estimates that cover ranges described for all present and past populations of anatomically modern humans, with the

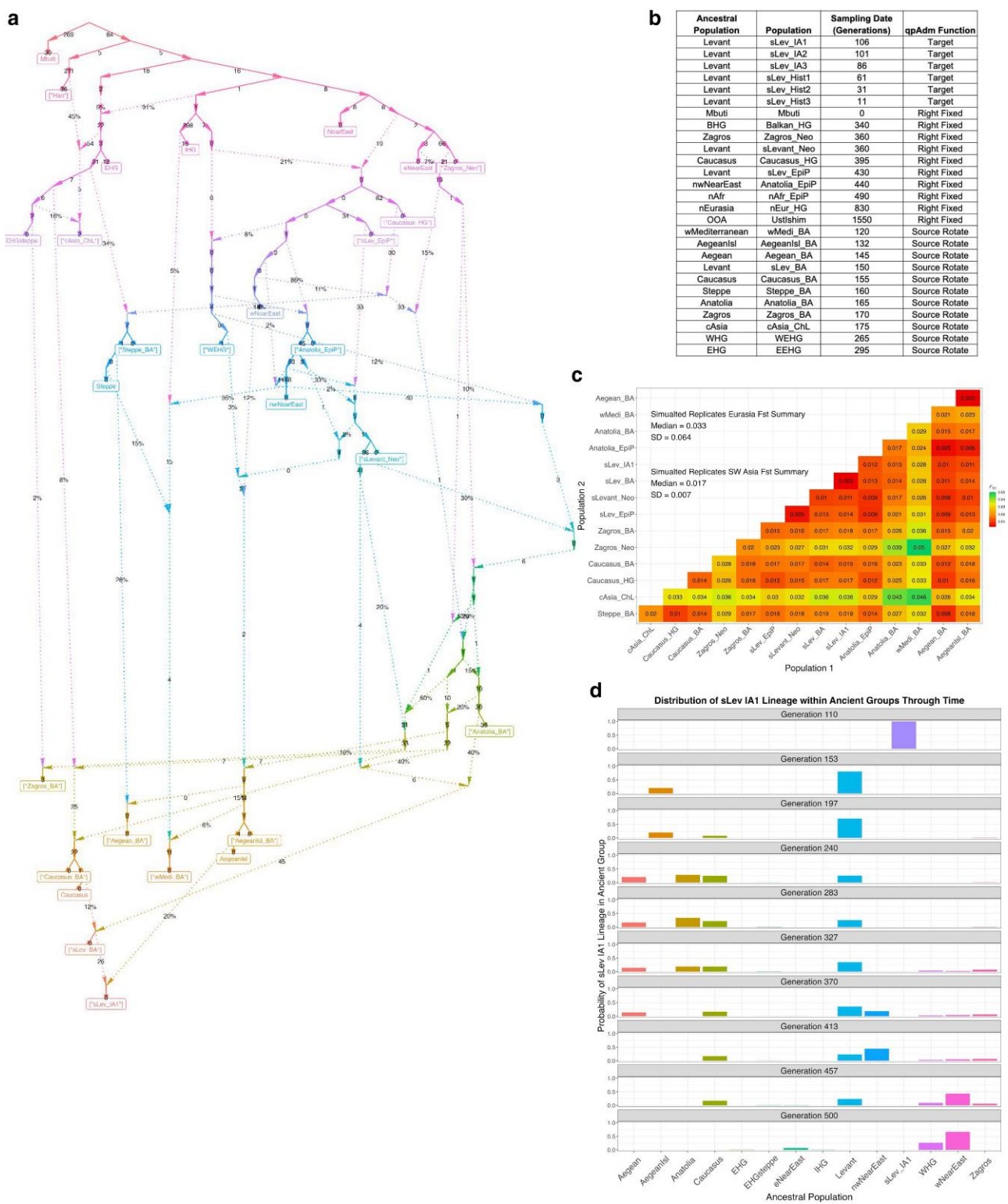

**Fig. 4.** a) Simplified topology of the simulated Eurasian demography. Square brackets indicate sampled populations. Pulse admixture proportions are indicated by a % value. Genetic drift values separating lineages are indicated by whole numbers and are computed as the number of generations separating nodes divided by the population size. b) A table of the sampled populations used in qpAdm analysis and the ancestral populations they split from (corresponding to ancestral populations in a). An $F_{ST}$ matrix c) for the sampled simulated populations is also shown. d) Barplots showing probabilities of encountering a lineage found in the "sLev IA1" group in other simulated ancestral populations (only presenting populations with nonzero probabilities). The ancestral populations are those from which we sampled and correspond to the first column in b).

median pairwise $F_{ST}$ between all source and right populations spanning from ~0.00012 to ~0.15. Due to extensive admixture between ancient groups occupying regions across Eurasia beginning around the 6th millennium BCE, populations from historical periods exhibit, on average, lower genetic differentiation than their predecessors (Fig. 1b). Therefore, evaluating archaeogenetic

hypotheses regarding historical migrations necessitates the ability to disentangle the admixture histories of minimally differentiated ancient groups separated by very short periods of genetic drift. To address this, we used demographic Model 1 (Fig. 3a) to directly evaluate the impact and limits of population differentiation on the performance of rotating qpAdm. In addition, for all downstream demographic inference analyses across all simple demographic models, we constrained the simulated parameter space to values that approximate conditions observed among historical period groups such as a low median pairwise $F_{ST}$ between 0 and 0.02 computed on the S1, S2, R1, R2, and R3 populations and $\leq$ 100 generations since the admixture event forming the target population.

## The limits of population differentiation for qpAdm admixture model inference

A requirement of qpAdm is that at least one right group population is differentially related to populations in the left set (Haak *et al.* 2015; Harney *et al.* 2021) as the power of qpAdm is in part due to the right group populations' ability to distinguish between putative ancestry sources (Harney *et al.* 2021). Consistent with this principle, we observe a general trend of increasing qpAdm performance (QTP) with larger median pairwise $F_{ST}$ values (Fig. 3e). As these values approach ~0.009, equivalent to empirical values observed among Bronze Age and older groups occupying Europe, the Mediterranean, and Southwest Asia, we notice QTP to asymptote around ~0.8 and convergence on an average of two plausible qpAdm models per simulation iteration (Fig. 3e). However, as the median pairwise $F_{ST}$ drops to values observed at the lower ends of human population differentiation (~0.003), we observe a sharp decline in the average QTP driven by increases in both the number of plausible false qpAdm models and rejections of the true qpAdm model (S1 + S2; Fig. 3e).

To analyze the distribution of plausible qpAdm models driving the QTP variation at different levels of genetic differentiation, we formed median pairwise $F_{ST}$ bins roughly corresponding to values separating historical epochs. The smallest range, $F_{ST}$ between 0 and 0.008, corresponds to the diversity estimated from samples dating between 1,500 and 3,200 years ago (Fig. 1b) with the upper range broadly demarcating the Iron Age from the Bronze Age in Southwest Asia. The middle range, $F_{ST}$ between 0.008 and 0.013, corresponds to the diversity estimated from samples dating between 3,200 and 5,500 years ago and encompasses the Bronze Age population diversity (Fig. 1b). The upper range, $F_{ST}$ between 0.013 and 0.02, estimated from samples dating between 5,500 and 8,500 years ago, represents the diversity present among populations ancestral to those of the historical period (Fig. 1b). Consistent with the QTP distribution described above, the smallest $F_{ST}$ bin contains the highest number of false-plausible qpAdm models, including single-source models for the target population (Fig. 5a). The degree of population divergence also impacts the plausibility of the true model with larger $F_{ST}$ bins increasing both the frequency of plausible true models (0.705, 0.859, and 0.842 for the three $F_{ST}$ bins, respectively) and the proportion of true models out of all plausible qpAdm models (22.5%, 44.8%, and 48.7%, for the three $F_{ST}$ bins, respectively). Notably, the increased rejection of the true model in the lowest $F_{ST}$ bin is largely due to inaccurate estimations of the admixture weights. Approximately 50% of the true model replicates with $P$-values $\geq$ 0.05 are rejected due to admixture proportions outside the [0,1] range (Supplementary Fig. 2). For larger $F_{ST}$ bins, the predominant rejection of the true model shifts to statistical

significance, as the majority of true model replicates are rejected because of $P$-values between 0.01 and 0.05 (Supplementary Fig. 2).

With the recent shift of aDNA research toward reconstructing admixture histories within subcontinental regions (Ávila-Arcos *et al.* 2023), understanding the limits of rejecting false sources recently split from the true ancestral source is becoming increasingly pertinent. To investigate this, we explored the limits of differentiating between the sister clades of R1 and S1, and by symmetry S2 and R2 (Fig. 3a), as false and true sources in qpAdm models. As expected, qpAdm has the greatest difficulty rejecting models that combine one of the (false source) cladal populations with one of the true sources, as combinations of S1 + R2 and S2 + R1 account for more than 25% of all plausible qpAdm models across all $F_{ST}$ bins (Fig. 5a). As anticipated given the topological symmetry of Model 1, the two false qpAdm models are plausible at almost equal frequency. However, we observe less frequently both false models plausible within the same simulation (Supplementary Fig. 3), consistent with the convergence toward an average of two plausible qpAdm models described above (Fig. 3e).

To assess the relationship between genetic differentiation within putative source clades and performance of rotating qpAdm, we analyzed the joint distribution of S1 + R1 and S2 + R2 $F_{ST}$ values for all false qpAdm models that included one of the R1 or R2 populations. As expected, we observe on average larger $F_{ST}$ values between the S1 + R1 and S2 + R2 populations for rejected false models (mean = 0.004 and median = 0.002) than plausible false models (mean = 0.002 and median = 0.001) which resulted in statistically significant differences between their respective $F_{ST}$ distributions (Mann–Whitney $U$ $P$-value < 0.001; Supplementary Fig. 4). Thus, our simulations suggest that under the simple topological structure of Model 1, rotating qpAdm has the power to differentiate between closely related cladal populations, albeit with more difficulty distinguishing between putative sources separated on the order of $F_{ST} < {\sim}0.002$.

## Admixed sources and qpAdm demographic inference

Often, complex ancestral relationships exist among putative historical source populations [e.g. Lazaridis *et al.* (2016)]; however, whether this is detrimental to the effectiveness of identifying admixture patterns through qpAdm remains unknown. To assess the impact of the introduction, phylogenetic origin, and number of admixture events into the source population on admixture inference, we performed 5,000 simulations on each of the three simple demographic models that introduce admixture to the source (S1) population (Fig. 3b–d), with all other simulation parameters remaining consistent with Model 1 (Fig. 3a).

Importantly, the addition of admixture events into the S1 population does not lead to significant changes to the distribution of QTP across the $F_{ST}$ range. The results for all simple demographic models converge on a maximum average QTP of ~0.8 and an average of two plausible qpAdm models (Fig. 3e–h). We do observe subtle differences in their average performance for metrics such as false positive rate (FPR) = FP/(FP + TN), false discovery rate (FDR) = FP/(FP + TP), QTP, and QTP-binary (Table 1). From each simulation iteration, we computed the qpAdm FPR for each demographic model as follows: we counted the number of plausible false qpAdm models [false positives (FP)] to obtain the FP qpAdm model count. To obtain the number of true negative (TN) qpAdm models, we counted the number of rejected false qpAdm models. For example, in Model 1 simulation iteration 1,998, we have an FPR of 0.8 that occurred because, of the 21 total single- and two-source

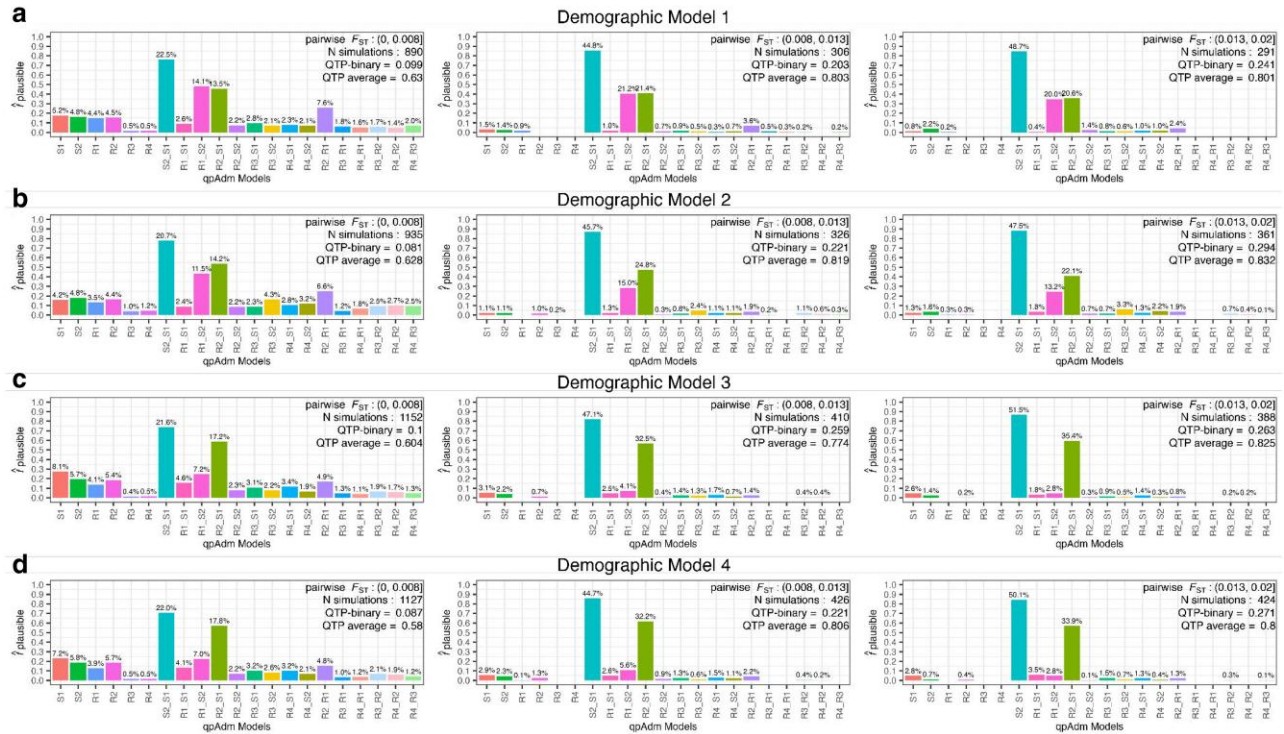

**Fig. 5.** Distribution of plausible single-source and two-source qpAdm models across three population differentiation ranges (between the S1, S2, R1, R2, and R3 populations) and across the four simple demographic models. The number of generations since admixture is less than or equal to 100 in all cases. Each row a–d represents one simulated demographic history, and the columns are increasing ranges of population differentiation ($F_{ST}$) corresponding to the historical period demarcations indicated in Fig. 1b. The values above each barplot represent the proportion each qpAdm model is plausible among all plausible qpAdm models within the simulation iterations for each differentiation range. The y-axis shows the frequency of each model as plausible across the total number of simulations within each differentiation range. In the top right corner of each barplot is the $F_{ST}$ range, number of simulations within that range, and the average QTP and QTP-binary.

qpAdm models, we have 16 FP qpAdm models and four false qpAdm models were rejected (FP/(FP + TN) = 16/20). We computed the FDR in the same fashion. The observation of a plausible true qpAdm model (S1 + S2) represents the true positive (TP) count, meaning an FDR of 1 occurs when only false qpAdm models are plausible and 0 when only the TP qpAdm model is observed and all false qpAdm models are rejected. We then averaged these metrics to generate a summary of the overall performance for each model under historical period parameters. No single demographic model consistently outperforms others across all performance metrics, indicating that different admixture scenarios have varying effects on qpAdm performance and accuracy. This is further supported by the observation that across multiple model plausibility criteria (discussed further below), the average QTP, QTP-binary, and FDR consistently favor demographic Model 2, while the FPR is most frequently lowest for Model 4 (Table 1). However, we note that the best-performing average qpAdm metric consistently falls within one of the more complex Models 2–4, suggesting that, on average, the introduction of admixture to the source population increases qpAdm rotation performance even though it decreases overall population differentiation (both median and average $F_{ST}$ is largest in Model 1).

Similarly to Model 1, all demographic models with complex admixture history of S1 exhibited the highest number of false-plausible qpAdm models in the lowest range of population divergence (Fig. 5). However, while we observed very similar frequencies of plausible S1 + R2 and S2 + R1 false qpAdm models under demographic Model 1, the introduction of admixture to the S1 population introduced an asymmetry, with the S2 + R1 qpAdm

model being more frequently rejected than the S1 + R2 model (Fig. 5b–d). Interestingly, this asymmetry is most pronounced under Models 3 and 4, which includes admixture from the common ancestor of S2 and R2 (iS2R2) to the S1 branch, and the asymmetry further increases in larger $F_{ST}$ bins (Fig. 5c and d). Demographic Model 4 including two admixture events in S1 displayed a distribution of false-plausible models across sources intermediate between Models 2 and 3 (including one admixture event in S1) suggesting that the phylogenetic source of gene flow in S1 has a greater impact on the resulting plausible qpAdm models than the number of admixture events in S1 (Fig. 5b–d). This has important implications for the empirical study of ancient populations whose sources are themselves admixed (see Discussion).

A further challenge in the use of aDNA in resolving hypotheses regarding migrations during historical periods is the increased likelihood of studying recent admixture events. Such scenarios may arise in the context of detecting shifts in genetic ancestry after episodes of human migration, where only a few generations separate the timing of admixture and the ancient human individuals sampled. Moreover, the effectiveness of qpAdm in addressing historical questions that necessitate the identification of a specific population or lineage responsible for admixture is inversely proportional to the number of plausible models it identifies. We assessed the performance of qpAdm under both of these challenges by modeling the interaction between generations since admixture and population divergence on the probability of exclusively identifying the true qpAdm model using a logistic generalized additive model in the mgcv v.1.9.0 R package (Wood 2004) with QTP-binary as the response variable, automatic

**Table 1.** Performance summaries of qpAdm rotation analysis for the four demographic models and different performance metrics.

| Plausibility criteria | Model 1 FPR | Model 1 FDR | Model 1 QTP | Model 1 QTP-binary | Model 2 FPR | Model 2 FDR | Model 2 QTP | Model 2 QTP-binary | Model 3 FPR | Model 3 FDR | Model 3 QTP | Model 3 QTP-binary | Model 4 FPR | Model 4 FDR | Model 4 QTP | Model 4 QTP-binary |
|---|---|---|---|---|---|---|---|---|---|---|---|---|---|---|---|---|
| P-value 0.01 | 0.3178 | 0.8443 | 0.6439 | 0.0000 | 0.3014 | 0.8063 | 0.6585 | 0.0031 | 0.3298 | 0.8417 | 0.6389 | 0.0000 | 0.3209 | 0.8387 | 0.6467 | 0.0010 |
| P-value 0.05 | 0.2753 | 0.8407 | 0.5983 | 0.0000 | 0.2549 | 0.7946 | 0.6273 | 0.0086 | 0.2823 | 0.8297 | 0.6238 | 0.0041 | 0.2742 | 0.8293 | 0.6161 | 0.0040 |
| P-value 0.01 + weights [0,1] | 0.1185 | 0.5890 | 0.7611 | 0.1210 | 0.1283 | 0.5921 | 0.7619 | 0.1295 | 0.1179 | 0.5943 | 0.7185 | 0.1251 | 0.1115 | 0.5826 | 0.7241 | 0.1371 |
| P-value 0.05 + weights [0,1] | 0.0983 | 0.5737 | 0.6993 | 0.1479 | 0.1072 | 0.5744 | 0.7122 | 0.1566 | 0.0966 | 0.5658 | 0.6834 | 0.1656 | 0.0918 | 0.5659 | 0.6761 | 0.1553 |
| P-value 0.01 + weights [0,1] ± 2 SE | 0.0743 | 0.6244 | 0.6076 | 0.1170 | 0.0805 | 0.6103 | 0.6396 | 0.1307 | 0.0675 | 0.6444 | 0.5213 | 0.1159 | 0.0653 | 0.6282 | 0.5372 | 0.1275 |
| P-value 0.05 + weights [0,1] ± 2 SE | 0.0601 | 0.6097 | 0.5539 | 0.1426 | 0.0662 | 0.5924 | 0.5966 | 0.1591 | 0.0541 | 0.6135 | 0.4952 | 0.1523 | 0.0523 | 0.6146 | 0.4955 | 0.1421 |
| All single-source models rejected | | | | | | | | | | | | | | | | |
| P-value 0.01 + weights [0,1] | 0.0699 | 0.6312 | — | — | 0.0739 | 0.6306 | — | — | 0.0653 | 0.6504 | — | — | 0.0639 | 0.6296 | — | — |
| P-value 0.05 + weights [0,1] | 0.0600 | 0.6087 | — | — | 0.0652 | 0.6051 | — | — | 0.0555 | 0.6111 | — | — | 0.0548 | 0.6057 | — | — |
| P-value 0.01 + weights [0,1] ± 2 SE | 0.0675 | 0.6420 | — | — | 0.0701 | 0.6303 | — | — | 0.0620 | 0.6630 | — | — | 0.0604 | 0.6427 | — | — |
| P-value 0.05 + weights [0,1] ± 2 SE | 0.0565 | 0.6223 | — | — | 0.0605 | 0.6062 | — | — | 0.0512 | 0.6251 | — | — | 0.0503 | 0.6225 | — | — |
| All single-source models rejected and significant $f_3$-statistics | | | | | | | | | | | | | | | | |
| P-value 0.01 + weights [0,1] | 0.0524 | 0.6182 | — | — | 0.0559 | 0.6012 | — | — | 0.0530 | 0.6554 | — | — | 0.0514 | 0.6330 | — | — |
| P-value 0.05 + weights [0,1] | 0.0427 | 0.5950 | — | — | 0.0454 | 0.5736 | — | — | 0.0428 | 0.6247 | — | — | 0.0416 | 0.6083 | — | — |
| P-value 0.01 + weights [0,1] ± 2 SE | 0.0523 | 0.6252 | — | — | 0.0554 | 0.6039 | — | — | 0.0525 | 0.6635 | — | — | 0.0506 | 0.6425 | — | — |
| P-value 0.05 + weights [0,1] ± 2 SE | 0.0425 | 0.6030 | — | — | 0.0450 | 0.5764 | — | — | 0.0422 | 0.6341 | — | — | 0.0408 | 0.6208 | — | — |

Each cell contains the average of each performance metric under different qpAdm plausibility criteria. The averages are over the parameter range of the historical period (admixture generations ≤100 and median pairwise $F_{ST} > 0$ and ≤0.02). The performance metrics are as follows: FPR, false positive rate; FDR, false discovery rate; QTP, qpAdm test performance; and QTP-binary, qpAdm test performance provided that only the true model fits the data. Each performance metric is evaluated under a different model plausibility criterion for the four demographic models. Their averages are printed in each cell.

smooth terms for each of the predictor variables (median pairwise $F_{ST}$ between the S1, S2, R1, R2, and R3 populations; generations since admixture), and the model parameters were estimated using restricted maximum likelihood. The model's output was in the form of log-odds, which we then converted to probabilities. This conversion was done by first exponentiating the log-odds to get the odds ratio and then dividing the odds ratio by 1 plus the odds ratio [i.e. probability of QTP-binary = odds ratio/(1 + odds ratio)]. We visualized these predicted probabilities on a grid that represents the space of the historical parameters (Fig. 3i–l).

As expected, larger median pairwise $F_{ST}$ values resulted in increased QTP-binary probability for all simple demographic models (Fig. 3i–l), with the more complex Models 2–4 performing better than Model 1 across historical $F_{ST}$ ranges. Counterintuitively, the model with admixture from the internal branch ancestral to both the S2 and R2 populations (iS2R2; Model 3) performed the best at $F_{ST}$ values both below (median pairwise $F_{ST} < 0.008$) and within (median pairwise $0.008 < F_{ST} < 0.013$) ranges approximating that of historical periods (Fig. 3i–l). When median divergence levels reached those of populations older than the Bronze Age (median pairwise $F_{ST} > 0.013$), the model with a gene flow from an out-group to a source branch (Model 2) outperformed the others, achieving the same QTP-binary probability values with fewer generations since admixture than Models 3 and 4 (Fig. 3j). It also had the highest maximum QTP-binary probability of all models, achieving this with generations since admixture greater than ~90 (Fig. 3j). In the absence of admixture events in the history of S1 (Model 1), we observed no significant impact of generations since admixture on the QTP-binary probability ($\chi^2 = 2.25$ and $P$-value = 0.089). However, all three admixed-source models, especially Model 3, show a weak but statistically significant effect of generations since admixture on QTP-binary, with the effect appearing more pronounced for larger $F_{ST}$ values (approximate significance of $T_{admix}$ predictor variable smooth term: Model 2 $\chi^2 = 9.94$ and $P$-value = 0.0012; Model 3 $\chi^2 = 23.79$ and $P$-value < 0.001; and Model 4 $\chi^2 = 10.17$ and $P$-value ≤ 0.001; Fig. 5i–l). The observed weak influence of generations postadmixture on the QTP-binary probability is likely a consequence of correlations between the $T_{admix}$ and $T_3/T_4$ parameters (Supplementary Fig. 1b) rather than a decline in performance due to more recent admixture. In support of this idea is that both Models 3 and 4, which incorporate admixture from the iS2R2 branch that is delineated by the $T_2$ and $T_4$ split-times (Fig. 3c, d), have the strongest correlation between $T_{admix}$ and $T_4$ of all demographic models (Supplementary Fig. 1b). Conversely, under Model 2, $T_{admix}$ has the strongest correlation with parameter $T_3$ (Supplementary Fig. 1b), which determines the divergence time between R1 and S1.

## Accuracy of qpAdm admixture estimates

We also evaluated if the introduction of admixture to the ancestral source population (S1) would introduce bias or increase uncertainty in the admixture weight estimation of the target for the true qpAdm model. Consistent with previous studies (Harney et al. 2021), in the absence of ancestral admixture, we observed a delta-alpha (simulated minus estimated admixture weight) mean of 0 under Model 1 and an $R^2$ of 0.86 demonstrating that qpAdm can accurately estimate the simulated admixture weight without bias (one-sample $T$-test $P$-value = 0.65; Fig. 6). However, in the presence of admixture to the source population from an out-group, we observe a subtle overestimation of the S1 contribution to the target (Model 2, delta-alpha mean = 0.01, one-sample $T$-test $P$-value = 0.02). However, when admixture to S1 is from the iS2R2 branch, we observe an underestimation, albeit not significant, of

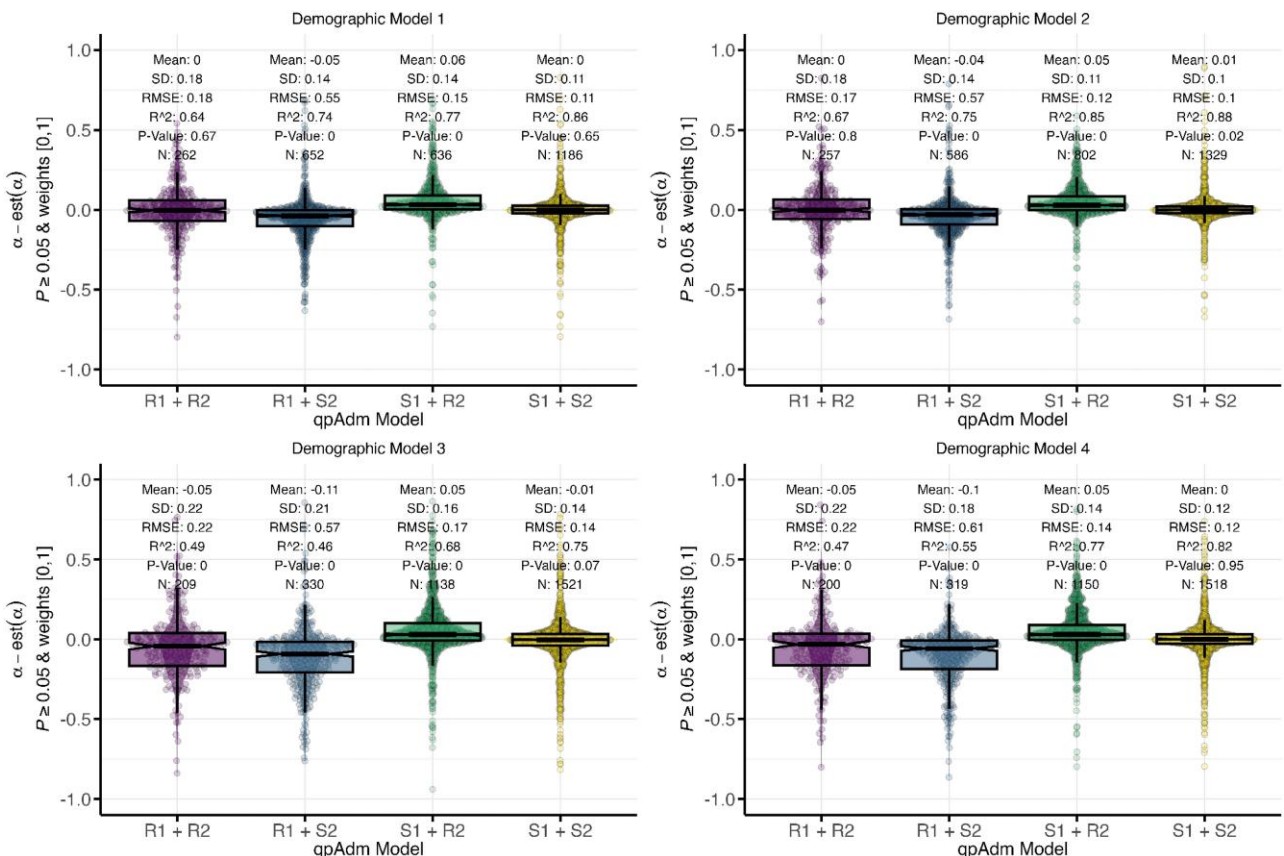

**Fig. 6.** Deviations of estimated from simulated admixture proportions for the R1 and S1 sources in the *qpAdm* models S1 + S2, R1 + R2, R1 + S2, and S1 + R2. Median pairwise $F_{ST}$ between the S1, S2, R1, R2, and R3 populations is between 0 and 0.02, and the number of generations since admixture is less than or equal to 100. Each panel shows results for one simple demographic model.

almost equal magnitude (Model 3, delta-alpha mean = −0.01, one-sample *T*-test *P*-value = 0.07; Fig. 6). The symmetrical biases between Model 2 and Model 3 appear to cancel out under demographic Model 4, where we observe a delta-alpha mean of 0 (one-sample *T*-test *P*-value = 0.95; Fig. 6). All demographic models exhibited similar levels of uncertainty in their admixture weight estimation (Fig. 6). While Model 3 performed the worst with the lowest $R^2$, largest delta-alpha standard deviation, and root mean square error (Fig. 6), the weight estimate uncertainty is considerably smaller than expected under completely random sampling (the SD of the difference between two uniformly distributed and uncorrelated random variables is 0.408) further supporting the accuracy of qpAdm admixture estimates under these conditions.

In empirical aDNA studies, one will often include multiple closely related populations in the qpAdm candidate source list to determine which is the best representation of the target ancestry. Therefore, we assessed how the selection of false sources and their phylogenetic relationship to the true source affected the bias and uncertainty in admixture weight estimates. Under the simplest model (Model 1), we observed that misspecified (false) models that combine the true source populations with the sister clade of the other true source (R1 + S2 or S1 + R2) resulted in an almost equal overestimation of the simulated admixture weight for the true source (R1 + S2 mean = −0.05, *T*-test *P*-value ≤ 0.001 whereas S1 + R2 mean = 0.06, *T*-test *P*-value < 0.001; Fig. 6). However, when both sources are equally phylogenetically distant from the true admixing sources (R1 + R2), we only observed an increase in the weight estimate uncertainty but no bias (SD = 0.18 and *T*-test *P*-value = 0.67; Fig. 6). We observed similar qualitative patterns

in the admixed-source models (Models 2–4), with a bias in overestimating the contribution from the true source when paired with one of the false sources (S1 + R2 and S2 + R1 *T*-test *P*-values < 0.001; Fig. 6). Interestingly, the largest effects are observed under demographic Models 3 and 4 for the qpAdm model S2 + R1, suggesting that admixture from the internal iS2R2 branch to the ancestral S1 branch increases the overestimation of the S2 contribution to the target (Fig. 6). Moreover, selecting the two symmetrical populations R1 + R2 results in an overestimation of the R2 contribution under both Models 3 and 4 (*T*-test *P*-value < 0.001), but we observed no bias under Model 1 (*T*-test *P*-value = 0.67). Additionally, the model with a gene flow from an out-group to the ancestral S1 branch (Model 2) has no bias in admixture weight estimation (R1 + R2 *T*-test *P*-value = 0.8), further supporting the impact of the admixture between ancestral source branches (iS2R2) on the qpAdm weight bias (Fig. 6).

## Impact of block jackknife size under recent admixture

In order to quantify the uncertainty linked to the random sampling of SNPs, qpAdm employs a jackknife resampling method to calculate standard errors. Using simple demographic Model 1, we investigated how altering the block jackknife size affects the performance of qpAdm, specifically in cases of recent admixture by testing eight different block sizes ranging from 0.01 to 100 Mb across three generations since admixture bins [$T_{admix}$ ranges = (0, 50), (50, 100), and (100, max)]. We find our results are consistent with Harney *et al.* (2021), who evaluated the impacts of block jackknife sizes on qpAdm performance from a fixed demography.

Our analysis shows that qpAdm estimates of admixture proportion remain unbiased, regardless of the block size, for both recent [$T_{admix} = (0, 50)$] and older [$T_{admix} = (50, 100)$ and $(100, max)$] generations since admixture (Supplementary Fig. 12). Additionally, our findings show that the smallest block sizes result in the lowest standard error estimates (Supplementary Fig. 11b). However, we observe a slight increase in the standard error across all block sizes under recent admixture (i.e. <50 generations), which suggests that the use of standard errors as a qpAdm plausibility constraint might negatively impact performance under recent admixture (discussed further below). Consistent with the findings of Harney *et al.* (2021), the smallest and largest block sizes yield nonuniformly distributed *P*-values, and we do not observe changes to the *P*-value distribution across the three generations since admixture bins (Supplementary Fig. 11a).

## qpAdm plausibility criteria and improving model inference accuracy

A number of different qpAdm plausibility criteria are employed in empirical aDNA analysis such as *P*-value thresholds of 0.01 (e.g. Skoglund *et al.* 2017; Narasimhan *et al.* 2019; Bergström *et al.* 2022; Lazaridis *et al.* 2022b; Koptekin *et al.* 2023; Skourtanioti *et al.* 2023) and 0.05 (e.g. Olalde *et al.* 2019; Sirak *et al.* 2021; Salazar *et al.* 2023), the use of ±2-SE constraint on the admixture weights (Narasimhan *et al.* 2019), the requirement of the rejection of all single-source models, and favoring simpler models over more complex ones (Lazaridis *et al.* 2016, 2022a; Skoglund *et al.* 2017; Narasimhan *et al.* 2019; Salazar *et al.* 2023). Our objective was to determine how these plausibility criteria impact the performance and accuracy of qpAdm admixture model inference. Additionally, we aimed to assess whether the accuracy of qpAdm could be improved by conditioning on a significant admixture $f_3$-statistic for plausible two-source models, a method developed to test if a target population is consistent with being formed from two putative sources (Patterson *et al.* 2012; Peter 2016, 2022).

We observed a substantial decrease in the average error rates (FPR and FDR, as defined above) and an increase in average performance metrics (QTP and QTP-binary) across all demographic models when introducing the admixture weight [0,1] constraint to the plausibility criteria in qpAdm (Table 1). We note that the admixture weight [0,1] constraint is the most common additional constraint on qpAdm model plausibility in the aDNA literature. However, adding the additional ±2-SE weight constraint, while reducing the FPR for all demographic models, also increased the FDR for both *P*-value thresholds (Table 1), highlighting the trade-off between rejecting the true model and failing to reject false models when assessing accuracy. Similarly, the ±2-SE weight constraint also decreased the average QTP results for both *P*-value thresholds across all models, and only Model 2 shows an increase in average QTP-binary (increases in QTP-binary for both *P*-values 0.01 and 0.05; Table 1).

We also evaluated the impact of requiring all single-source qpAdm models to be rejected on the FPR and FDR error rates. Under this condition, we updated our FPR calculation as follows: for each simulation iteration, if at least one false single-source qpAdm model was plausible, all two-source qpAdm models were rejected and we then computed the FPR as the FP/(FP + TN) following the guide above. Meaning, a two-source qpAdm model can only contribute to the FPR if all single-source qpAdm models are rejected in its simulation iteration. Recalling the above example, in the demographic Model 1 simulation iteration 1,998, we had an FPR of 0.8 that occurred because, of the 21 total single- and two-source qpAdm models, we have 16 FP models, six of which are single-source models. However, because we now condition on all single-source models to be rejected, we have six false positives (single-source models) and 14 true negatives (rejected two-source). As such, conditioning on the rejection of all single-source qpAdm models in this instance results in a reduction in the FPR from 0.8 to 0.3. The FDR was computed following the same procedure, where all two-source qpAdm models are rejected if a single-source model is plausible in their simulation iteration. Importantly, we found that requiring all single-source models to be rejected increased the FDR for all demographic models at both *P*-value thresholds (Table 1). Conversely, we find that the FPR is decreased with the rejection of all single-source models for all demographic Models across all plausibility criteria (Table 1).

In addition to rejecting all single-source qpAdm models, the further criterion of a significant admixture $f_3$-statistic for plausible two-source qpAdm models resulted in the lowest error rates (FPR and FDR) for all demographic models (Table 1). The relationship between the power of the admixture $f_3$-statistic and demographic parameters was explored by Peter (2016). They showed through mathematical formulae [see equation 1 ( EQ:1)] and simple simulations similar to our Model 1 that the conditions of a negative $f_3$-statistic required a large number of generations between the split of the admixing sources ($T_2$) and the time of admixture ($T_{admix}$), a low probability of lineages in the target population coalescing before the admixture event ($T_{admix}$), and the admixture proportion ($\alpha$) close to 50%. As such, for any pair of true source populations to produce a negative $f_3$-statistic for a target, the demographic model from which they descend must conform to EQ:1:

$$\text{Negative } f_3\text{-statistic condition} = \left( \frac{1}{(1 - c_x)} \frac{T_{admix}}{T_2} < 2\alpha(1 - \alpha) \right) \quad (1)$$

where $c_x$ corresponds to the probability two lineages sampled in the target population that have a common ancestor before the time of admixture (Peter 2016).

By conditioning on simulations whose demographic parameters result in a negative $f_3$-statistic condition [i.e. the value of EQ:1 left-hand side (LHS) must be less than the value of the right-hand side (RHS)], we show that demographic models with admixture to the source (S1) population from the iS2R2 branch (Models 3 and 4) result in the largest $f_3$-statistic type II error rate (percent of simulations with $f_3$(target; S1, S2) Z-score > −3 for Models 1, 2, 3, and, 4 are 34%, 30%, 44%, and 43%, respectively; Supplementary Fig. 5a). As such, we show that the models with a gene flow from the iS2R2 branch require, on average, a larger LHS to RHS difference (smaller ratio of LHS/RHS) for the negative $f_3$-statistic condition to generate significance [median EQ:1 LHS/RHS ratio for $f_3$(target; S1, S2) Z-scores < −3 across Models 1, 2, 3, and, 4 are 0.158, 0.128, 0.085, and 0.078, respectively]. Given a substantial period of independent drift between the ancestral split of the sources and the time of admixture is a prominent factor in $f_3$-statistic negativity, this power reduction appears to be principally driven by admixture between the ancestral source lineages decreasing the amount of independent genetic drift separating $T_{admix}$ and $T_2$. Importantly, all the demographic effects on $f_3$-statistics power described above are magnified when selecting the wrong source pairs (Supplementary Fig. 5b–d).

## qpAdm model ranking by *P*-value

A common application of qpAdm, and by extension qpWave, is ranking model performance via *P*-values (van de Loosdrecht *et al.* 2018; Lazaridis *et al.* 2022a; Oliveira *et al.* 2022; Moots *et al.* 2023; Taylor *et al.* 2023). We evaluated the use of *P*-values for the relative ranking of qpAdm models by assessing how frequently

each of the single- and two-source qpAdm models had the largest, second-largest, third-largest, and fourth-largest $P$-values for each of the 5,000 simulations. Across all demographic models, the true qpAdm model significantly outperformed all other qpAdm models by having the largest $P$-value in more than 60% of the simulations (Supplementary Table 1a–d). Additionally, we found that both the relative ranking and frequency of $P$-values reflected the underlying demography and frequency of plausible models described above (Fig. 5). Under Model 1, the S1 + R2 and S2 + R1 models had the largest $P$-value with about equal frequency (0.127 and 0.137, respectively), whereas under demographic Models 2–4, the S1 + R2 qpAdm model has the largest $P$-value approximately 10× more frequently and is the second best performing of all qpAdm models (Supplementary Table 1a–d).

## Going complex: evaluating the use of aDNA in reconstructing admixture histories under complex human demography

To better reflect empirical aDNA conditions, we expanded our admixture inference evaluation to encompass a complex Eurasian demographic model (for details, refer to Complex demographic model: parameter configurations under Materials and methods). Although not replicating true human history, this framework facilitates the evaluation of admixture inference constraints in complex demographic contexts using aDNA. Our simulations resulted in expected levels of population divergence given empirical observations with a median pairwise $F_{ST}$ of 0.03 between all ancient groups representing Eurasian populations and 0.017 among those representing Southwest Asian populations (Fig. 1b). A pairwise $F_{ST}$ matrix computed on the first replicate shows expected genetic affinities among the analysis populations (Fig. 4c). We used the tskit lineage_probabilities() function to further assess the relationship between the target and analysis populations by tracking the location of lineages sampled from the target through time among the remaining simulated demographic lineages (Fig. 4d). The results show that between the youngest and oldest populations included in the qpAdm analysis, lineages from the target population are principally found in the Levant, Aegean, AegeanIsl, Anatolia, and Caucasus ancestral groups (Fig. 4d).

## qpAdm performance and aDNA data quality

Consistent with the results previously shown by Harney *et al.* (2021), the degree of data missingness appears to be one of the primary factors influencing the performance of rotating qpAdm. Below, we adopt the term "coverage" to represent the proportion of SNPs nonmissing. Thus, the aDNA missingness sampling condition of SNPs < 100,000 represents the lowest-coverage data set, the random missingness sampling condition represents the medium-coverage data set, and the missingness condition of 100,000 < SNPs < 500,000 represents the highest-coverage data set. The lowest-coverage data set produced the largest frequency of plausible single-source and two-source qpAdm models (Fig. 7), resulting in the lowest average QTP, largest FPR and FDR, and an average QTP-binary of 0 (Supplementary Table 3). The two higher-coverage aDNA sampled data sets resulted in very similar qpAdm performance with the highest-coverage data set performing slightly better than the medium-coverage data set as it both rejects all single-source qpAdm models and has less total plausible qpAdm models (Fig. 7), resulting in on average higher QTP and QTP-binary and the lowest FPR and FDR (Supplementary Table 3). Since the degree of missingness in the AADR random sampling scheme sometimes results in populations with more missingness than the lowest-coverage data set (Supplementary

Table 2), the relative performance of these missing data schemes highlights the importance of maximizing data coverage in all populations, not just the target, for rejecting false qpAdm models.

Interestingly, we note that among the two higher-coverage data sets, the plausible false qpAdm models are not arbitrarily selected as they descend from ancestral populations that are shown above to harbor the target population lineages (Fig. 4a and d). The single exception to this pattern is the simulated wMedi_BA population which, when paired with the sLev_BA population, is plausible at 10% in each of the higher-coverage data sets and 76% in the lowest-coverage data set (Fig. 7). The simulated wMediterranean lineage, from which wMedi_BA descends, is modeled as receiving 6% admixture from the true source population 29 generations before the formation of the target population (Fig. 4a; Supplementary File 1). This demonstrates that correlations in allele frequencies between populations driven by admixture from a shared ancestral source, in addition to shared inherited genetic drift, can result in false positive qpAdm results.

Interestingly, the rotating qpAdm analysis on the simulated whole-genome branch-length $f_2$-statistic rejected all false qpAdm models and classified the true model plausible in 70% of the replicates (Fig. 7). We ran a receiver operating characteristic (ROC) curve analysis where we varied the $P$-value between 0 and 1 to assess the relationship between $P$-value thresholds and qpAdm performance as measured by the TPR and FPR. In calculating the ROC, we constrained the qpAdm models to have plausible admixture weights [0,1] and performed each calculation on 3,000 $P$-value thresholds between 0 and 1. These results revealed that for the whole-genome branch-length $f_2$-statistic data set, the qpAdm TPR converges to 100% with $P$-values greater than 1 × $10^{-3}$ and the FPR does not increase until the $P$-value reaches 0 (Supplementary Fig. 6). All data sets with aDNA missingness exhibit a trade-off of covarying increases/decreases in the FPR/TPR with changes to the $P$-value threshold (Supplementary Fig. 6), which we also observe in the distribution of $P$-values for qpAdm models with plausible admixture weights (Supplementary Fig. 7). Importantly, both higher-coverage aDNA data sets have greater than 89% TPR and less than 1% FPR with a $P$-value threshold of 0.1, suggesting an additional strategy for increasing qpAdm accuracy (Supplementary Fig. 6).

## qpAdm model ranking by $P$-value in aDNA under complex demography

We evaluated the accuracy of determining the best-fitting qpAdm model by ranking them by their $P$-values given the admixture complexity of the simulated Eurasian demography. Under the higher-coverage data sets, the true model has the largest $P$-value in more than 90% of the simulation replicates and has the largest $P$-value in 100% of the replicates using the highest-coverage data set (Supplementary Table 4). However, caution should be applied to ranking qpAdm by $P$-values in data sets with low coverage as in our lowest aDNA coverage data set, the true model has the largest $P$-value in only 16% of the replicates, second to the false model of sLev_BA + Anatolia_BA (Supplementary Table 4). The observation of the sLev_BA + Anatolia_BA and sLev_BA + Aegean_BA source combinations as the alternative qpAdm models that possessed the largest qpAdm $P$-value in at least one replicate demonstrates the difficulty in rejecting closely related candidate sources ($F_{ST}$ between the AegeanIsl_BA or Aegean_BA and Anatolia_BA populations ~0.003 and 0.017, respectively). Nonetheless, the sLev_BA population is consistently paired with alternative sources in the most frequent qpAdm models with the largest $P$-values, suggesting that the identification of

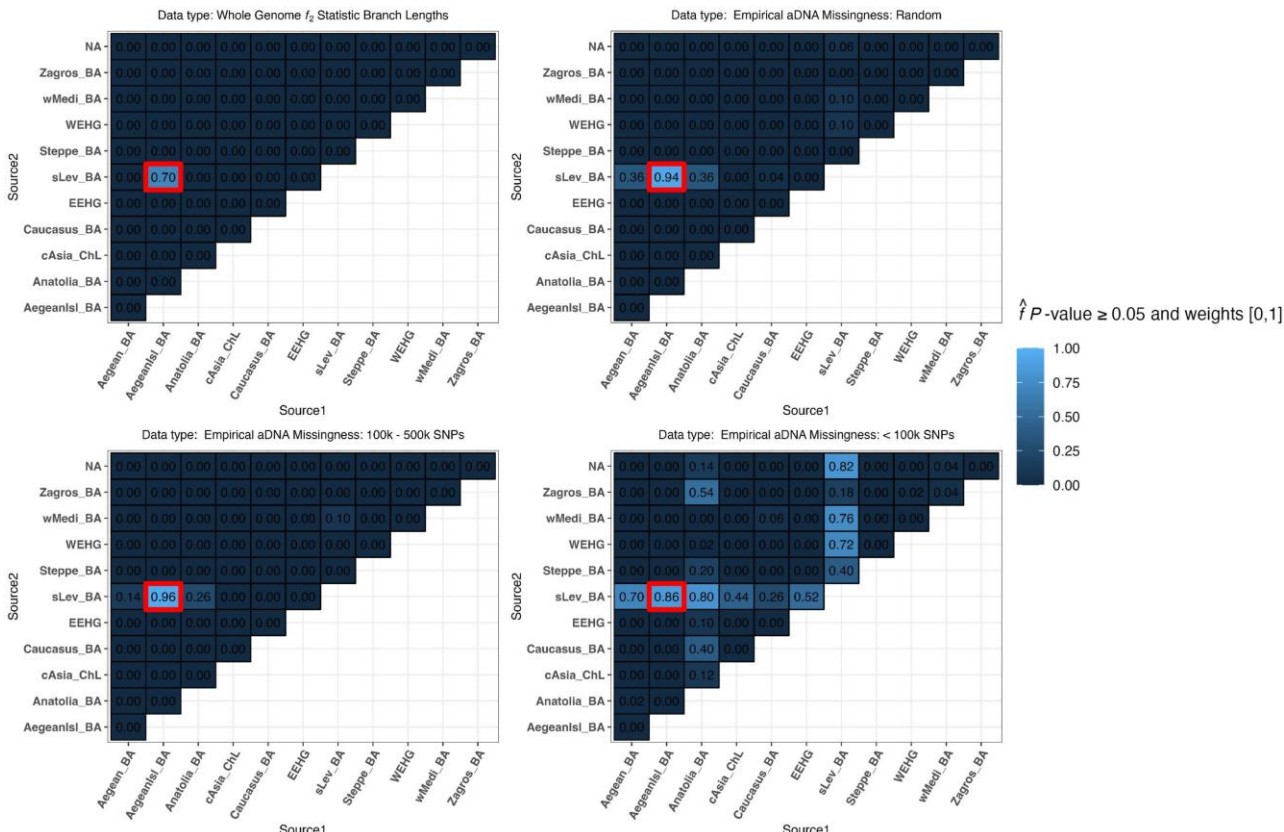

**Fig. 7.** Heatmaps of the proportion of replicates with plausible P-value ≥ 0.05 and weights [0,1] for the complex demography (Aegean Island admixture to Southern Levant) for the target population "sLev IA1." The thick outlined red box represents the most optimal true model. The results are presented for four data sets: $f_2$-statistics calculated on whole-genome branch lengths and the three data sets with varying SNP missing rates.

overrepresented populations in high-ranking qpAdm models is a suitable heuristic to determine one of the likely true sources regardless of the degree of data missingness (Supplementary Table 4).

## Generations since admixture and qpAdm performance and accuracy

We also explored the effect of post-admixture drift on qpAdm performance. Importantly, across all descendent target populations and degrees of data missingness, we observe no significant trend in qpAdm performance or admixture weight accuracy (Supplementary Fig. 8). Under the whole-genome branch-length $f_2$-statistic data set, the admixture weight SE appears to increase with increasing generations since admixture; however, it has no significant impact on the accuracy or precision of the admixture weight estimates (Supplementary Fig. 8). As expected, we observe the largest estimated admixture weight SE and delta-alpha values in the lowest-coverage data set, with between 0.12 and 0.20 SD on delta-alpha (Supplementary Fig. 8). As such, caution should be given to interpreting admixture proportions from data sets of low coverage.

## qpAdm plausibility criteria

We evaluated the impact of the different qpAdm plausibility criteria described in the simple demography section on our complex demographic aDNA simulations. In contrast to the simple demographic simulations, the introduction of the ±2-SE constraint on admixture weight estimates consistently either reduced or maintained the FPR for all aDNA missingness conditions and either reduced or maintained the FDR in all but the lowest-coverage data

sets (Supplementary Table 3). Of note is that each aDNA missingness data set has a different plausibility criterion that maximizes its QTP, making the selection of single plausibility criteria to maximize QTP infeasible. We do, however, observe for all data sets, the lowest error rates (FDR and FDR) with the cocriteria of rejection of all single-source models, P-value ≥ 0.05, and ±2-SE constraint on the admixture weight estimates, albeit with greater than 0.98 FDR in the lowest-coverage data set (Supplementary Table 3). The plausibility criteria of P-value ≥ 0.05 and admixture weights [0,1] result in the smallest FDR in the lowest-coverage data set. In empirical studies with low-coverage aDNA samples, this may represent the most optimal plausibility criteria as it minimizes the frequency of type II errors as evidenced by the largest QTP value (Supplementary Table 3). The distribution of P-values for models with plausible admixture weights [0,1] (Supplementary Fig. 7) and the ROC curve analysis (Supplementary Fig. 6) show that increasing the P-value threshold for the low-coverage data set does not result in a significant reduction in the FPR without penalizing the TPR (Supplementary Fig. 6).

Importantly, we observe no impact from the use of significant admixture $f_3$-statistics as an additional plausibility criterion to increase qpAdm model inference accuracy as all pairwise combinations of qpAdm sources were not significant regardless of data quality (Supplementary Fig. 9). As described above, the power for the detection of admixture from $f_3$-statistics is strongly influenced by the underlying population demography and divergence of the candidate sources from the true admixing populations. Our simple demography simulations showed that both gene flow between source lineages and increased divergence of the

candidate source population from the true admixing source reduced the $f_3$-statistics power. Under the Eurasian demography, the two source groups, Levant and Aegean, undergo recent bidirectional gene flow after the split from their most recent common ancestral population. We computed the $f_3$-statistics negativity condition (EQ:1) for both the split-time of the Levant and Aegean sources and the date of admixture for all target populations. As expected, we observe an increase in $f_3$-statistic estimates with increasing generations since admixture (Supplementary Fig. 9). Moreover, the estimated $f_3$-statistic negativity appears to conform closer to the $f_3$-statistics negativity condition (EQ:1) when computed using the date of most recent bidirectional admixture between the Levant and Aegean sources rather than their split-time (Supplementary Fig. 9). This further supports the idea that admixture between source lineages negatively impacts the power of the admixture $f_3$-statistic and highlights the importance of utilizing $f_3$-statistic estimates as confirming plausible qpAdm models rather than rejecting false models, similar to how they were originally proposed (Patterson et al. 2012).

## Discussion

The qpAdm software has become one of the hallmark methods in archaeogenetic analyses for reconstructing admixture histories of ancient populations (see Lazaridis et al. 2016, 2022a, 2022b; Skoglund et al. 2017; Harney et al. 2018; Mathieson et al. 2018; Antonio et al. 2019; Narasimhan et al. 2019; Fernandes et al. 2020; Marcus et al. 2020; Ning et al. 2020; Wang et al. 2020, 2021; Yang et al. 2020; Carlhoff et al. 2021; Librado et al. 2021; Papac et al. 2021; Sirak et al. 2021; Bergström et al. 2022; Maróti et al. 2022; Patterson et al. 2022; Changmai, Jaisamut, et al. et al. 2022a, Changmai, Pinhasi, et al., 2022b; Lee et al. 2023). This is due in part to its modest computational requirements, use of allele frequency data, and minimal model assumptions (Haak et al. 2015; Harney et al. 2021). The primary motivation for this work is addressing its applicability, performance, and limits in reconstructing admixture histories under challenging scenarios that emerge when reconstructing population dynamics within the historical period. Such conditions range from identifying the true source population among minimally differentiated candidates and potential biases that may arise from sources that are admixed and ancestrally connected through complex demographies. It also may involve dealing with short intervals between the admixture event of interest and the ancient sample. Additionally, we sought to determine how these challenges are impacted by missing data typical of aDNA conditions.

We addressed these questions through simulations of both simple admixture-graph–like demographies that explore a broad parameter space on two chromosomes and whole-genome simulations of admixture-graph–like demography that reflects the inferred complexity of Eurasian population history. We note that computing qpAdm on chromosomes 1 and 2 subsetted from the 50 whole-genome complex demography simulations exhibit standard errors within the range observed under the aDNA 100,000–500,000 SNP and random sampling missingness approaches (Supplementary Fig. 13), demonstrating the relevance of our inferences from the simple demographic simulations for empirical aDNA analysis. Moreover, this highlights the complementarity of our approach, enabling us to simultaneously investigate a broad parameter space that is computationally infeasible under whole-genome simulations—while evaluating the effects of aDNA data missingness and the upper boundaries of qpAdm performance that would otherwise be obscured using smaller genomes.

It is important to acknowledge that our study configures all demographies as a series of discrete population splits and pulse admixture events, each separated by periods of independent genetic drift (that is why these simulations are termed "admixture-graph–like"). Thus, if the distribution of human settlements across the ancient landscape aligns more closely with temporally evolving stepping-stone models, the interpretations drawn from our study may lose some of their significance. Also of note is that all of our demographic models adhere to the fundamental assumptions of qpAdm (Harney et al. 2021): (1) there are no gene flows connecting lineages private to candidate source populations (after their divergence from the true admixing populations) and "right group" populations and (2) there are no gene flows from the fully formed target lineage to "right group" populations (Harney et al. 2021). It is crucial to recognize that these assumptions might be frequently violated when investigating demographic history in the historical period and beyond it, leading to false rejections of true simple models. In turn, these prompt researchers to test more complex models which often satisfy qpAdm model plausibility criteria but are misleading when subjected to historical interpretation (Yüncü et al. 2023). If stringent sampling criteria, as outlined in our companion paper (Yüncü et al. 2023), are not diligently followed, these violations are shown to pose substantial challenges to the effective use of qpAdm in demographic inference.

Our simple demographic simulation results show that qpAdm converges on its maximum QTP as the median pairwise $F_{ST}$ of the sample set approaches ~0.005–0.008 (Fig. 3e–h), well within the diversity expected of historical period populations (Fig. 1b). However, we find a much larger level of population divergence is required, with a median pairwise $F_{ST}$ exceeding 0.015, to simultaneously reject all false models and identify the true model with a probability greater than 30% (Fig. 3i–l). This finding suggests that highly specific archaeogenetic hypotheses that require the sole identification of the correct model may currently lie beyond the capabilities of qpAdm given the prevailing data conditions. Importantly however, within the set of models considered plausible by qpAdm under both the simple admixture simulations and Eurasian whole-genome complex simulations, we consistently observe that one of the true sources is included in those most frequently accepted models, irrespective of the degree of population divergence or levels of data missingness (Figs. 5a–d and 7).

When it comes to distinguishing between closely related cladal populations, such as the differentiation between S1 and R1 or S2 and R2 in our simple admixture simulations, our results suggest that qpAdm exhibits heightened discriminatory power when these closely related cladal populations have diverged on the order of $F_{ST} >$ ~0.002–0.004 (Supplementary Fig. 4). A similar result emerges from our complex Eurasian demographic simulations. For instance, the candidate source Aegean_BA, which is modeled as having recently split from the true source AegeanIsl_BA, is differentiated at a median $F_{ST}$ of 0.003 and is frequently included in plausible qpAdm models at all levels of data missingness (Fig. 7). However, it's worth noting that in the complex Eurasian demographic simulations, population divergence alone does not exclusively determine the probability of a false source appearing in a plausible qpAdm model. For instance, we frequently observe the Anatolian_BA population in plausible qpAdm models (Fig. 7) while it is both approximately equally divergent from the true sources (median $F_{ST}$ Aegean_BA = 0.016 and sLev_BA = 0.015; Fig. 4c). This is likely driven by demographic factors analogous to the conditions of simple demographic Models 3 and 4 (Fig. 3c and d) whereby the Eurasian demographic model includes bidirectional gene flow between the ancestral Levant and Anatolian

populations (30% Levant to Anatolia and 40% Anatolia to the Levant in generations 305 and 224, respectively) resulting in a substantial likelihood that lineages from the target population are present within the Anatolian population (Fig. 4d).

A key discovery with relevance for archaeogenetic research in regions with complex migration histories is that introducing admixture into the source population (but not violating the topological assumptions described above) can notably improve qpAdm's performance, especially when considering conditions that resemble the typical levels of divergence observed during the historical period. Notably, the phylogenetic origin of the ancestral admixture differently impacts qpAdm accuracy (FPR and FDR) and performance (QTP). For instance, when the gene flow originates from an out-group to the target, true sources, and candidate source populations, it yields the highest average QTP performance and lowest FDR among all demographic models. In contrast, we observe lower FPR under demographic models that include admixture between sources (Model 3) than from an out-group (Model 2) and the lowest FPR when both ancestral source admixture events occur (Model 4; Table 1). Overall, this trend appears to be primarily driven by the increased differential relatedness between left and right set qpAdm populations, irrespective of the decrease in average population divergence.

This observation is consistent with theoretical expectations regarding the way qpAdm uses *P*-values to reject candidate models (Haak *et al.* 2015; Harney *et al.* 2021). When the target and right group populations share genetic drift distinct from the shared ancestry between the target and the putative left group sources, this will result in the rejection of the left group sources as an admixture model of the target population given a certain *P*-value threshold. As such, the ancestry inherited by the target from a source that is itself admixed increases the number of populations that it uniquely shares drift with. This is evident in the increased power to reject the false R1 + S2 qpAdm models with the introduction of admixture to the S1 ancestral source lineage (Fig. 5a–d). Consequently, these observations underscore the importance in empirical aDNA studies of prescreening qpAdm right groups to optimize genetic differentiation and differential relatedness with potential source populations for maximizing qpAdm performance, as originally proposed in Haak *et al.* (2015).

We also note that so long as the correct source populations are chosen, usage of source populations with complex admixture history does not introduce bias in the estimation of admixture weights (Fig. 6). However, we do observe a reduction in accuracy when admixture to a source lineage occurs from another source branch, in contrast to an out-group (Fig. 6). Most notably, the selection of source populations appears to be more critical for accurately estimating admixture contributions. We observed a significant bias toward the population closest to the true source, leading to an overestimation of admixture proportions for this population (Fig. 6). This phenomenon is present in all simple demographic models but appears to be more pronounced in models with ancestral admixture to the source (Fig. 6). In cases where both populations are equidistant from the true admixing sources, the bias is only evident in models that include an admixture event between source branches (Fig. 6).

We have observed that two additional criteria significantly enhance the accuracy of qpAdm model inference. The first involves considering two-source (admixture) qpAdm models only when all single-source models are rejected. The second criterion involves deeming these models as plausible when the source pairs generate a significantly negative admixture $f_3$-statistic (Z-score < −3). While these criteria have proven effective in reducing bias (FPR

and FDR) across a wide range of demographic parameters, in empirical studies, it is crucial to assess the anticipated parameters of each demographic model being evaluated before applying these criteria universally. This is because certain demographic conditions can increase the FDR. For example, we observe an increase of plausible single-source models when the admixture weight is close to 1 (Supplementary Fig. 10), which, if requiring all single-source models to be rejected, will result in the more frequent false rejection of the true model.

As for the criterion of a significant admixture $f_3$-statistic, under conditions where there is only a short period between the split of the admixing source populations and their admixture to form the target and when the admixture proportions deviate significantly from 0.5, the power of the $f_3$-statistic to detect an admixture event diminishes, increasing type II errors (Supplementary Fig. 5). We observe this scenario in our Eurasian demographic simulations where admixture between the ancestral sources after their split decreased the power of the admixture $f_3$-statistic, resulting in a 100% type II error rate (Supplementary Fig. 9). Moreover, the simple simulations reveal that this effect is exacerbated by the divergence between the tested candidate population and the true admixing source, making the conditions for negativity of the $f_3$-statistic even more stringent (Supplementary Fig. 5). Therefore, we suggest that the $f_3$-statistic should be used as a confirmation and ranking tool for plausible qpAdm models, rather than as a criterion for rejecting them (i.e. favoritism is given to plausible models with a significant $f_3$-statistic over those without). This aligns with the original interpretative guidance when using the $f_3$-statistic as a formal admixture test (Patterson *et al.* 2012).

With respect to the procedure of ranking feasible qpAdm models based on their *P*-values, the initial suggestions from Harney et al. (2021) raised a notable concern and advised against *P*-value ranking to identify the best model. Their argument is based on the observation that *P*-values under false models closely related to the true model are almost uniformly distributed and that in cases when multiple models are plausible, any one false model could easily have a larger *P*-value than the true model (Harney *et al.* 2021). However, our findings from simple admixture demographic simulations show that in over 60% of the simulations, the true model has the largest *P*-value (Supplementary Table 1). Similarly, in the complex Eurasian simulations under the condition of high coverage, we found that in over 90% of the replications, the true model had the largest *P*-value (Supplementary Table 4). In both simple admixture and complex Eurasian simulations, we also found that the relative ranking of *P*-values accurately reflected how closely a false model represented the true ancestry of the target population. Therefore, provided that an empirical data set has good coverage, we propose that ranking qpAdm models by *P*-values can offer valuable additional information for determining the true model.

A surprising finding from the complex Eurasian demographic simulations was when performing the qpAdm rotation analysis with $f_2$-statistics computed from the whole-genome branch lengths, we obtained a 100% true positive rate and a 0% FPR with a *P*-value threshold of 0.001 (Supplementary Fig. 6). This outcome underscores the remarkable potential inherent in the principles underlying qpAdm, while also highlighting the constraints imposed by data scale. In light of this, we suggest that there is room for significant enhancements in the qpAdm protocol from methods that can leverage more accurate estimations of $f_2$-statistics. Possible avenues for this improvement could be developing innovative techniques for extracting information from ancestral recombination graphs within the context of aDNA, or conditioning on the site frequency spectrum for the enrichment

of rare alleles. As such, it is clear from these results that further improvements would make qpAdm a powerful tool for accurately reconstructing the genetic histories of populations under the most complex scenarios.

## Data availability

The authors affirm that all data necessary for confirming the conclusions of the article are present within the article, figures, tables, and Supplementary material. Both the simple demographic and complex Eurasian Model simulations were written in Snakemake pipelines to facilitate reproducibility and can be accessed via our GitHub repository (https://github.com/archgen/complex_demog_sims). Supplemental figures and tables are available in the supplemental material.

Supplemental material available at GENETICS online.

## Acknowledgments

We are grateful for the advice and helpful suggestions from Yassine Souilmi, Raymond Tober, and Bastien Llamas at an initial stage of the project. All simulations were performed on the computational infrastructure at the Pennsylvania State University's Institute for Computational and Data Sciences' Roar supercomputer. We note that our content is solely the responsibility of the authors and does not necessarily represent the views of the Institute for Computational and Data Sciences. Empirical aDNA data was obtained from the Allen Ancient DNA Resource v.52.2 (Mallick et al. 2024).

## Funding

C.D.H. and M.P.W were funded by the National Institute of Health under award number R35GM146886. P.F. was supported by the Czech Science Foundation (project no. 21-27624S led by P.F.); the Ministerstvo Školství, Mládeže a Tělovýchovy (program ERC CZ, project no. LL2103); the John Templeton Foundation (grant no. 61220 to David Reich), a gift from Jean-Francois Clin, and the financial support of the European Union under the LERCO project number CZ.10.03.01/00/22_003/0000003 via the Operational Programme Just Transition.

## Conflicts of interest

The authors declare no conflicts of interest.

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

*Editor: S. Ramachandran*