## [Peer Review File · Genetics]

Testing Times: Disentangling Admixture Histories in Recent and Complex Demographies using ancient DNA

Matthew Williams, Pavel Flegontov, Robert Maier, and Christian Huber

NOTE: The reviews and decision letters are unedited and appear as submitted by the reviewers.

In extremely rare instances and as determined by a Senior Editor or the EIC, portions of a review may be redacted. If a review is signed, the reviewer has agreed to no longer remain anonymous.

The review history appears in chronological order.

Review Timeline:

Submission Date:	2023-11-01
Editorial Decision:	2024-02-09
Resubmission Received:	2024-04-08
Accepted:	2024-06-11

February 8, 2024

GENETICS-2023-306605

Testing Times: Challenges in Disentangling Admixture Histories in Recent and Complex Demographies

Dear Dr. Huber:

An expert in the field has reviewed your manuscript, and I have read it as well and discussed it with multiple colleagues on the GENETICS editorial board. We appreciate this study of the behavior of two popular methods for inference of admixture histories, qpAdm and f3-statistics, and the exploration of their behavior across different classes of complex models for aDNA and other applications. While your manuscript is not currently acceptable for publication in GENETICS, we would welcome a substantially revised manuscript that addresses the reviewer's comments. You can read their review at the end of this email.

We look forward to receiving your revised manuscript. Please let the editorial office know approximately how long you expect to need for revisions.

Upon resubmission, please include:

1. A clean version of your manuscript;
2. A marked version of your manuscript in which you highlight significant revisions carried out in response to the major points raised by the editor/reviewers (track changes is acceptable if preferred);
3. A detailed response to the editor's/reviewers' feedback and to the concerns listed above. Please reference line numbers in this response to aid the editor and reviewers.

Your paper may be sent back out for review.

Additionally, please ensure that your resubmission is formatted for GENETICS

<https://academic.oup.com/genetics/pages/general-instructions>

Follow this link to submit the revised manuscript: Link Not Available

Sincerely,

Sohini Ramachandran
Associate Editor
GENETICS

Approved by:
Nicholas Barton
Senior Editor
GENETICS

Reviewer #1 (Comments for the Authors (Required)):

[I want to disclose that I also reviewed the companion qpAdm study by Yüncü et al. which is under revision at the same time. Given that both papers share co-authors and have thus likely been worked on in parallel, some of my comments will apply to both papers and might be present in both of my reviews.]

In the presented manuscript, the authors provide an evaluation of the behavior of two popular methods for inference of admixture histories, qpAdm and f3-statistics. They first use simulations of four minimal demographic models which capture important aspects of admixture histories (including increases in complexity in histories of candidate sources), and later move to investigating qpAdm performance in conditions more resembling the empirical reality in terms of model complexity and data quality. Looking at the qpAdm inference accuracy as a metric of differentiation of source populations through the lens of Fst is particularly interesting and relevant, as the aDNA field attempts to study populations with increasingly entangled histories within increasingly narrowly defined geographic regions and time periods. The qpAdm tool has become very popular across the field of ancient DNA yet, quite surprisingly, its behavior across different classes of complex models has become virtually unexplored. The presented manuscript aims to do fix this problem and provides an important evaluation of the limits and power of qpAdm from a rigorous population genetic/statistical perspective. I am sure the manuscript will be of a huge service to the entire aDNA community and will provide an important set of decision guidelines in future qpAdm applications.

I don't have major complaints about the methodology or statistical design of the manuscript. That said, I do have several comments which I believe should be addressed in order to make the conclusions and lessons presented by the manuscript clearer to the aDNA readership and easier to put into the context of empirical aDNA work and into practical use.

Major comments

=====

lines 90-91: "We started by simulating two chromosomes of combined length ~ 491 Mbp under four simplistic and qualitatively different admixture graphs"

Why have the authors simulated only two chromosomes of 491 Mb in total when in the second stage, for the complex model, they do simulate truly genome-scale data (so computational cost isn't the issue)?

My question is motivated by this: If I simulate replicates of a 1Mb, 10Mb, 100Mb chromosome and compute (for instance) an f_4 statistic to test an admixture hypothesis, I will get f_4 values which will be less noisy the more sequence is simulated (and more accurate admixture inferences). I would expect the same should apply even for qpAdm, with an increase in power with larger sequence lengths, purely by reducing statistical noise?

In order to make the results more comparable to the behavior of real data (and between the two simulation studies in the manuscript), wouldn't it be more accurate to simulate replicates from a single simulation of all 22 autosomes for each topology (about 3000 Mb genomes)? It's not immediately obvious to me whether the f -statistics/qpAdm noisiness expected in real data would be otherwise comparable to these 491 Mb simulations.

I find this decision even more surprising because on line 95 the authors write that they simulated truly whole-genome simulations for their complex models. So why not make the simulation setups the same across both simulation studies?

caption of Figure 1B: Why have the authors opted to focus on F_{st} values specifically in ancient Southwest Asia? Given the interesting results showing the relationship between qpAdm power and population differentiation throughout the manuscript, why not show F_{st} from the entire AADR panel? In fact, this could be even partitioned into panels showing this style of age-vs- F_{st} plots across time periods in different geographic regions (Europe, Southwest Asia, etc.?). This would make it much easier for a reader to put results such as those in Figure 3 into a wider context, including Europe where majority of aDNA is still coming from.

On a related note, while referring to the same Figure 1B on lines 225-227 the authors talk about "ancient southwest Eurasian groups". Then again, line 241 talks again about "Southwest Asians". This makes me think it should be one or the other in all of those instances.

lines 242-243: "convergence on an average of two plausible qpAdm models per simulation iteration".

Reading this made me again think about the effect of the length of the genome simulated, and whether simulating sequence spanning all autosomes (and not just 491 Mb) as mentioned in my comment above would change the proportion of significant models. If the proportion of plausible models is expected to change with genome-scale data, I'm not sure how translatable the described results are towards running qpAdm on empirical data which is, of course, genome-wide.

lines 556-577: This paragraph describing a complex simulated model would be incomparably easier to read if it were accompanied by a population phylogenetic tree (or admixture graph) directly in the main text. After all, the model represents a second major case study in the manuscript. It would also make it immediately possible to compare this model to models in the first study shown in Figure 3 and see differences in topological complexity.

lines 182 and 617: The authors mention using 5 Mb block size for performing the jackknife.

How does the size of genome blocks used for jackknifing affect the results of a qpAdm analysis? If admixture is recent, ancestry haplotypes (and extend of linkage between admixed SNPs in a target) will be larger, so blocks overlapping those haplotypes will

be entirely formed by SNPs linked on those haplotypes. If admixture is ancient, ancestry haplotypes will be shorter, which will also show at the level of blocks (depending on the block sizes). As such, does the block size have any influence on qpAdm inferences, depending on the age of admixture being modelled? If there is such an effect, perhaps showing a simple figure demonstrating p-values and inferred admixture proportions as a function of block size would be useful. If there's not an effect and this is of no concern at all, a reference to the relevant literature would be helpful.

Given that both submitted qpAdm manuscript were apparently worked on in tandem, it would be useful for readers to know if this manuscript follows the advice from the bigger companion paper, in particular with respect to separating (or not) target/left/right sets based on sampling time, etc (i.e. advice that's stated in the "Best practices" list of Yüncü et al.). Stating the relationship of the simulation setup of this manuscript to the best practices recommended by the companion study would help to put results and lessons in this paper into the context of the other study.

The authors recommend f3 as a confirmation and ranking tool for plausible qpAdm models. The companion paper recommends using ADMIXTURE. Should both be used where applicable? How does increasing power with f3 help compared to ADMIXTURE and vice versa?

Minor comments

=====

lines 21-22: "under conditions resembling historical populations"

I assume "aDNA conditions" are meant here given the previous sentence, but those don't really apply to "populations". I suggest rephrasing this sentence to clarify what's intended to be said here.

line 171-173: A sentence or two describing why the two-phase simulation approach combining DTWF and standard coalescence is necessary might be helpful for readers, particularly if they were interested in doing their own specific simulation follow up analyses.

Figure 3: I find the text in the legends on the bottom of the figure a little confusing and it took me a while to decipher it just by looking at the figure:

I would suggest expanding the legend belonging to panels I-L a bit more, perhaps with an explicit/longer text such as "Prob. of QTP-binary (assuming qpAdm p-value ≥ 0.05 & admixture proportions [0, 1])". Or, if this would be deemed to long, something slightly more explicit because the current "(QTP - binary) $p \geq 0.05$ & weights [0:1]" is too terse.

I also suggest replacing "N: FP qpAdm models" with more conventional "# N of qpAdm models" in the equation using the convention of # to represent counts (if this is indeed what is intended by "N:").

line 178: Typo in the specified Ne value (10,00).

line 424: When the authors write "subtle and weakly significant", do they talk about effect size? Otherwise, at a given significance level, something either is or isn't significant; talking about "weak significance" seems vague. I wasn't sure how to interpret this statement. Clarifying it with more explicit and rigorous language would help.

line 464: The terms FPR and FDR appear to be first mentioned here but are not formally defined above in the context of admixture graphs. On line 481 a FPR definition is stated but it's not clear upon initial reading whether this definition is applicable to FPR discussed in the previous two paragraphs above this line.

Also, for formal reasons, the first instances of FPR and FDR on line 464 should be spelled out in their full non-abbreviated forms regardless of how obvious and generally known they are.

I was slightly surprised by seeing notation $[x:y]$ indicating that a number lies between x and y throughout the manuscript. Perhaps this is inspired by R vector syntax or something similar but why not use a more common notation for intervals in mathematics: $[x, y]$ for closed intervals and (x, y) for open intervals, and combinations of both for semi-closed intervals?

On a related note, the Fst-bin columns in Figure 4 share right or left Fst boundaries, respectively (0.008 and 0.013) but it's not clear which boundary is included in which panel. Again, using semi-closed interval notation will make this unambiguous.

Reply to Reviewer

Testing Times: Disentangling Admixture Histories in Recent and Complex Demographies using ancient DNA

Dear anonymous reviewer,

We sincerely thank you for your careful reading and review of our companion manuscripts. We believe your comments have substantially improved the translatability of our results for the ancient DNA (aDNA) community, in addition to strengthening a number of our inferences. In addressing your concerns, we have generated three additional supplementary figures and modified two main figures in accordance with your suggestions (see below).

We respond to each of the reviewers' comments separately, splitting each section with a horizontal line to increase readability. To further increase readability, we have enclosed the reviewers' comments with quotation marks ("..."), formatted our response text in **bold lettering font**, and included any manuscript text that was extensively modified in **blue color font** with associated line numbers L:X-Y taken from the updated manuscript.

Sincerely,

Matthew Williams on behalf of the authors.

"lines 90-91: "We started by simulating two chromosomes of combined length ~ 491 Mbp under four simplistic and qualitatively different admixture graphs". Why have the authors simulated only two chromosomes of 491 Mb in total when in the second stage, for the complex model, they do simulate truly genome-scale data (so computational cost isn't the issue)? My question is motivated by this: If I simulate replicates of a 1Mb, 10Mb, 100Mb chromosome and compute (for instance) an f_4 statistic to test an admixture hypothesis, I will get f_4 values which will be less noisy the more sequence is simulated (and more accurate admixture inferences). I would expect the same should apply even for qpAdm, with an increase in power with larger sequence lengths, purely by reducing statistical noise? In order to make the results more comparable to the behavior of real data (and between the two simulation studies in the manuscript), wouldn't it be more accurate to simulate replicates from a single simulation of all 22 autosomes for each topology (about 3000 Mb genomes)? It's not immediately obvious to me whether the f -statistics/qpAdm noisiness expected in real data would be otherwise comparable to these 491 Mb simulations. I find this decision even more surprising because on line 95 the authors write that they simulated truly whole-genome simulations for their complex models. So why not make the simulation setups the same across both simulation studies?"

We concur with the reviewer's intuition that 1) enhancements in qpAdm performance are likely to occur as the standard errors of the underlying f -statistics decrease, and 2) as genome sizes increase, the standard errors are expected to decrease. Furthermore, we acknowledge that our current manuscript does not effectively convey how our findings from the simple model, which uses 491Mb genome simulations, can be applied to "real data conditions" or our complex model simulations that incorporate whole-genome with aDNA conditions.

Firstly, while simulation software like msprime and the associated tree-sequence formats have significantly reduced the computational cost of simulations, executing human whole-genome simulations of approximately 3Gb remains quite resource-intensive. Consequently, we lack the resources to conduct 5,000 simulations for four simple models and were limited to 50 replicates of a single complex model. To enhance the clarity of our manuscript, we have added text detailing the computational cost of our whole-genome

simulations on lines 611-612 of the Results section for the complex simulations:

Our whole-genome simulations required an average of 12.6±2.2 GB of memory and a duration time of 221±10 hours per-replicate.

Additionally, we have added new text to the Discussion section (see below text from lines 805-814) that provides further clarity on the two approaches we implemented. It also highlights their synergistic relationship, which allowed us to evaluate admixture inference performance at two different scales of complexity.

To address the reviewers' concern regarding genome size and standard errors, we performed a subsampling of chromosomes 1 and 2 (approximately 491 Mb) from each of the 50 replicate whole-genome complex model simulations and executed qpAdm again. This re-analysis enabled us to compare the standard errors of qpAdm under the following five data quality conditions:

- 1. Branch length, whole genome.**
- 2. Branch length, chromosomes 1 and 2 (subsampled from the whole genome).**
- 3. aDNA, random sampling of ascertained SNPs (subsampled from the whole genome).**
- 4. aDNA, sampling of 100K to 500K ascertained SNPs (subsampled from the whole genome).**
- 5. aDNA, sampling of less than 100K ascertained SNPs (subsampled from the whole genome).**

In our revised Supplementary Figure 13 (below), we note that the standard errors calculated from the approximately 491 Mb genome are within the range of those derived from whole-genome simulations under both the random aDNA missingness sampling scheme, and when constraining the number of SNPs to be between 100K and 500K. Therefore, the insights gained from our simple simulations are directly applicable to analyses of empirical data. We have incorporated updated text into the Discussion section of the manuscript that explains these results and highlights the relevance of the simple simulations to the complex simulations and real-world data (see below text from lines 805-814).

Lines 805-814:

We addressed these questions through simulations of both simple admixture-graph-like demographies that explore a broad parameter space on two chromosomes, and whole-genome simulations of admixture-graph-like demography that reflects the inferred complexity of Eurasian population history. We note that computing qpAdm on chromosomes 1 and 2 subsetted from the 50 whole-genome simulations exhibit standard errors within the range observed under the aDNA 100k to 500k SNP and random sampling missingness approaches (SI Figure S13), demonstrating the relevance of our inferences from the simple demographic simulations for empirical aDNA analysis. Moreover, this highlights the complementarity of our approach, enabling us to simultaneously investigate a broad parameter space that is computationally infeasible under whole-genome simulations—whilst evaluating the effects of aDNA data missingness and the upper boundaries of qpAdm performance that would otherwise be obscured using smaller genomes.

Supplementary Figure S13

SI Figure S13. Evaluation of the qpAdm standard errors under varying simulated genome sizes and ancient DNA data missingness conditions using the complex simulation. The f2-statistic computed on chromosomes 1 and 2 was subsampled from the whole-genome simulations and qpAdm re-run to generate standard errors.

“Caption of Figure 1B: Why have the authors opted to focus on Fst values specifically in ancient Southwest Asia? Given the interesting results showing the relationship between qpAdm power and population differentiation throughout the manuscript, why not show Fst from the entire AADR panel? In fact, this could be even partitioned into panels showing this style of age-vs-Fst plots across time periods in different geographic regions (Europe, Southwest Asia, etc.). This would make it much easier for a reader to put results such as those in Figure 3 into a wider context, including Europe where majority of aDNA is still coming from.

On a related note, while referring to the same Figure 1B on lines 225-227 the authors talk about "ancient southwest Eurasian groups". Then again, line 241 talks again about "Southwest Asians". This makes me think it should be one or the other in all of those instances.”

We agree with the reviewer that incorporating more geographical regions will enhance the applicability of the analysis to a broader range of regions. Our focus was on ancient groups from southwest Asia, as this is the region that our complex demographic model is based on. We also acknowledge that while the number of published ancient genomes is growing rapidly, only a few regions have sufficient samples across all periods to conduct a temporal analysis of shifts in FST. Europe is one such region that the reviewer correctly identified. Therefore, we have included samples from southern-central-western Europe and the Mediterranean, in addition to the groups from southwest Asia, to illustrate the decrease in FST. Please refer to the updated Figure 1 below for the modifications.

Figure 1.

Dates of published aDNA samples. (A) A per-publication-year transect of the density of the (log₁₀) age of published ancient genomes. The publication dates and number of samples were taken from the Allen Ancient DNA Resource (AADR) v.52.2. (B) A temporal transect of population differentiation levels in southwest Asia, and Europe and the Mediterranean. The average dates for each sample in years BP were taken from the AADR v.52.2. For the plot in panel B, they were grouped into four epochs, with 3.2k years BP approximating the start of the Iron Age, 5.5k years BP approximating the start of the Bronze Age, 8.5k years BP approximating the start of the Neolithic period, and older years representing the Paleolithic period. The F_{ST} values were calculated using the Eigensoft v8.0.0 smartpca software.

In response to the author's inquiry into our use of the terms "Southwest Asians" (on original manuscript line 241) and "ancient southwest Eurasian groups" (on original manuscript lines 225-227), we thank the reviewer for their careful reading. We show below modified text from our updated manuscript where we had previously used the term "Southwest Asians" that describes below to provide more clarity for the reader in what we are referring to in each instance. We have now also made explicit reference to the group in question with respect to their geographical location.

Lines 228-230:

Due to extensive admixture between ancient groups occupying regions across Eurasia beginning around the 6th

millennium BCE, populations from historical periods exhibit, on average, lower genetic differentiation than their predecessors (Figure 1B).

Lines 245-246:

As these values approach 0.01, equivalent to empirical values observed amongst Bronze Age and older groups occupying Europe, the Mediterranean, and southwest Asia, ...

Lines 254-256:

The smallest range, F_{ST} between 0 and 0.008, corresponds to the diversity estimated from samples dating between 1.5k to 3.2k years ago (Figure 1B) with the upper range broadly demarcating the Iron Age from the Bronze Age in southwest Asia.

Lines 659-661:

Our simulations resulted in expected levels of population divergence given empirical observations with a median pairwise F_{ST} of 0.03 between all ancient groups representing Eurasian populations and 0.017 amongst those representing southwest Asian populations (Figure 6C).

“lines 242-243: "convergence on an average of two plausible qpAdm models per simulation iteration".

Reading this made me again think about the effect of the length of the genome simulated, and whether simulating sequence spanning all autosomes (and not just 491 Mb) as mentioned in my comment above would change the proportion of significant models. If the proportion of plausible models is expected to change with genome-scale data, I'm not sure how translatable the described results are towards running qpAdm on empirical data which is, of course, genome-wide.”

We believe the comment regarding the degree to which the ~490Mb simulations are transferable to ‘real empirical’ ancient DNA is adequately addressed in the first reply.

“lines 556-577: This paragraph describing a complex simulated model would be incomparably easier to read if it were accompanied by a population phylogenetic tree (or admixture graph) directly in the main text. After all, the model represents a second major case study in the manuscript. It would also make it immediately possible to compare this model to models in the first study shown in Figure 3 and see differences in topological complexity.”

We agree with the reviewer that the inclusion of a topology of the simulated demography would greatly enhance the interpretability of the model. However, we suggest that plotting such a complicated model becomes almost as un-interpretable as glancing at the simulation parameters table (Supplementary File SF1). As such, we have generated a simplified demography topology (Figure 6A) that is restricted to showing populations that are included in plausible qpAdm models, and ancestral populations that they share either from a shared admixture event or a shared ancestor.

Figure 6.

(A) Simplified topology of the simulated Eurasian demography. Square brackets indicate sampled populations. Pulse admixture proportions are indicated by a % value. Genetic drift values separating lineages are indicated by whole numbers and are computed as the number of generations separating nodes divided by the population size. (B) A table of the sampled populations used in qpAdm analysis and the ancestral populations they split from

(corresponding to ancestral populations in A). An FST matrix (C) for the sampled simulated populations is also shown. (D) Barplots showing probabilities of encountering a lineage found in the “sl.ev IA1” group in other simulated ancestral populations (only presenting populations with non-negative probabilities). The ancestral populations are those from which we sampled and correspond to the first column in B.

“How does the size of genome blocks used for jackknifing affect the results of a qpAdm analysis? If admixture is recent, ancestry haplotypes (and extend of linkage between admixed SNPs in a target) will be larger, so blocks overlapping those haplotypes will be entirely formed by SNPs linked on those haplotypes. If admixture is ancient, ancestry haplotypes will be shorter, which will also show at the level of blocks (depending on the block sizes). As such, does the block size have any influence on qpAdm inferences, depending on the age of admixture being modelled? If there is such an effect, perhaps showing a simple figure demonstrating p-values and inferred admixture proportions as a function of block size would be useful. If there's not an effect and this is of no concern at all, a reference to the relevant literature would be helpful.”

We appreciate the reviewer's question as it enables us to leverage our simulations under the Wright-Fisher (WF) model, which more accurately reflects long-range correlations across the genome due to recent admixture events. We conducted further analyses and incorporated the findings in a new section of the manuscript (see below). Our observations align with previous research on the performance of qpAdm under a simple demographic model by Harney et al. 2021. Moreover, we were able to show that the number of generations since admixture does not influence the distribution of p-values or the bias in admixture weights. Nevertheless, we did notice a slight increase in the standard error with a decrease in the number of generations since admixture, all of which we document below.

Lines 451-466:

Impact of block jackknife size under recent admixture

In order to quantify the uncertainty linked to the random sampling of SNPs, qpAdm employs a jackknife resampling method to calculate standard errors. Using simple demographic Model 1, we investigated how altering the block jackknife size affects the performance of qpAdm, specifically in cases of recent admixture by testing eight different block sizes ranging from 0.01 to 100 Mb across three generations since admixture bins (T_{adm} ranges = (0, 50), [50, 100), and [100, max)). We find our results are consistent with Harney et al. 2021, who evaluated the impacts of block jackknife sizes on qpAdm performance from a fixed demography. Our analysis shows that qpAdm estimates of admixture proportion remain unbiased, regardless of the block size, for both recent ($T_{\text{adm}} = (0, 50]$) and older ($T_{\text{adm}} = [50, 100)$, and [100, max)) generations since admixture (SI Figure S12). Additionally, our findings show that the smallest block sizes result in the lowest standard error estimates (refer to SI Figure S11B). However, we observe a slight increase in the standard error across all block sizes under recent admixture (i.e., < 50 generations), which suggests that the use of standard errors as a qpAdm plausibility constraint might negatively impact performance under recent admixture (discussed further below). Consistent with the findings of Harney et al., the smallest and largest block sizes yield non-uniformly distributed P-values, and we do not observe changes to the P-value distribution across the three generation since admixture bins (SI Figure S11A).

SI Figure S11. (A) Distribution of qpAdm P-values under simple demographic Model 1 for eight block sizes (0.01 to 100 Mb) referenced from Harney et al. and generations since admixture (T_{admix} ranges = (0, 50), [50, 100], and [100, max]). (B) Violin and boxplot plots of the qpAdm standard errors under the same jackknife block sizes and generations since admixture bins.

SI Figure S12. qpAdm weight estimate bias under simple demographic Model 1 for varying jackknife block size and generations since admixture. The bias is estimated under three temporal bins of generations since admixture (Tadmix ranges = (0, 50), [50, 100), and [100, max)) and eight block sizes (0.01 to 100 Mb) referenced from Harney et al. The bias ($\delta\alpha$) is measured as the difference between the simulated alpha and estimated alpha values. Histogram of the delta alpha is shown in a constrained range to visually account for outliers, and a scatterplot of the entire range is included within each jackknife block size and generation bin.

“Given that both submitted qpAdm manuscript were apparently worked on in tandem, it would be useful for readers to know if this manuscript follows the advice from the bigger companion paper, in particular with respect to separating (or not) target/left/right sets based on sampling time, etc (i.e. advice that’s stated in the “Best practices” list of Yüncü et al.). Stating the relationship of the simulation setup of this manuscript to the best practices recommended by the companion study would help to put results and lessons in this paper into the context of the other study.”

We appreciate the thorough analysis of both manuscripts by the reviewer. We have updated our manuscript to explicitly state that for the simple simulations, all the left and right populations are sampled simultaneously at a single “present” time period, i.e., generation 0.

Lines 136-137:

We note that all our simple demographic models described below do not violate the topological assumptions of qpAdm outlined in Harney et al. (Figure 3A-D) and all samples were taken at the “present”.

We have also revised our manuscript to clarify that our sampling under the complex demographic simulation model does not violate the assumptions of qpAdm and adheres to the best practices outlined in our companion paper, Yüncü et. al.

The following text has been added to the introduction of the complex model:

Lines 577-579:

We note that our sampling strategy for the simulated aDNA, whilst reflecting ages and sample sizes of empirical data, does not violate the qpAdm assumptions outlined in Harney et al. and discussed in our companion manuscript (see Discussion and Yüncü et al. 2023).

The following text has been added to the Discussion section:

Lines 820-824:

Also of note is that all of our demographic models adhere to the fundamental assumptions of qpAdm (Harney et al. 2021): 1) there are no gene flows connecting lineages private to candidate source populations (after their divergence from the true admixing populations) and "right-group" populations, and 2) there are no gene flows from the fully formed Target lineage to "right-group" populations (Harney et al. 2021).

The authors recommend f3 as a confirmation and ranking tool for plausible qpAdm models. The companion paper recommends using ADMIXTURE. Should both be used where applicable? How does increasing power with f3 help compared to ADMIXTURE and vice versa?

We thank the reviewer for their consideration of the implications of both papers in their review. While both the papers address the use of the qpAdm tool in modeling admixture histories of complex demographics, they have different objectives. Our companion paper by Yüncü et al. is designed to address qpAdm model violations, whereas our paper focuses on the power of qpAdm assuming no model violations.

In our paper, we delve into the demographic conditions under which admixture f3 statistics can be used as a validation tool for distinguishing between multiple plausible qpAdm models. This is discussed in detail in the section "qpAdm plausibility criteria" (lines 745-780). We highlight the demographic and sampling conditions that could increase type II errors and therefore suggest that the f3-statistic should be used more as a confirmation of a model's plausibility rather than a definitive test of its truth. For more guidance on this, please refer to our Discussion section (lines 789-941).

Regarding the use of ADMIXTURE, our companion paper does suggest that it can reduce the False Discovery Rate (FDR). However, we also caution that ADMIXTURE is not a formal model test and is subject to its own biases, as outlined in Lawson et al. 2018.

"lines 21-22: "under conditions resembling historical populations"

I assume "aDNA conditions" are meant here given the previous sentence, but those don't really apply to "populations". I suggest rephrasing this sentence to clarify what's intended to be said here."

We appreciate the reviewer's feedback on the clarity of our paper. We have subsequently revised the abstract to enhance its readability.

Lines 13-30:

Our knowledge of human evolutionary history has been greatly advanced by paleogenomics. Since the 2020s, the study of ancient DNA has increasingly focused on reconstructing the recent past. However, the accuracy of paleogenomic methods in resolving questions of historical and archaeological importance amidst the increased demographic complexity and decreased genetic differentiation remains an open question. We evaluated the performance and behavior of two commonly used methods, qpAdm and the f3-statistic, on admixture inference under a diversity of demographic models and data conditions. We performed two complementary simulation approaches –firstly exploring a wide demographic parameter space using branch-length data from two chromosomes under four simple demographic models of varying complexities and configurations– and secondly, we analyzed a model of Eurasian history composed of 59 populations using whole-genome data modified with ancient

DNA conditions such as SNP ascertainment, data missingness, and pseudo-haploidization. We observe population differentiation is the primary factor driving qpAdm performance. Notably, whilst complex gene-flow histories alter which models are classified as plausible, they do not reduce overall performance. Under conditions reflective of the historical period, qpAdm most frequently identifies the true model as plausible amongst a small candidate set of closely related populations. To increase the utility for resolving fine-scaled hypotheses, we provide a heuristic for further distinguishing between candidate models that incorporate qpAdm model P-values and f_3 -statistics. Finally, we demonstrate a significant performance increase for qpAdm using whole-genome branch-length f_2 -statistics, highlighting the potential for improved demographic inference that could be achieved with future advancements in f -statistic estimations.

“line 171-173: A sentence or two describing why the two-phase simulation approach combining DTWF and standard coalescence is necessary might be helpful for readers, particularly if they were interested in doing their own specific simulation follow up analyses.”

We appreciate the reviewer's feedback and have incorporated additional text to explain to the readers the rationale behind our two-phase simulation approach.

Lines 172-176:

To accurately capture the impacts of long-range linkage disequilibrium driven by recent admixture we used a two-phase process whereby for the first 100 generations into the past we simulated under the Discrete Time Wright-Fisher model (DTWF) (Nelson et al. 2020), and then under the Standard (Hudson) coalescent model until the most recent common ancestor (MRCA).

Lines 608-611:

As in the simple demographic model simulations, to capture long-range correlations across the genome due to recent admixture we simulated the first 25 generations into the past under the Discrete Time Wright-Fisher (DTWF) model (Nelson et al. 2020). We then simulated under the Standard (Hudson) coalescent model until the sequence MRCA.

“Figure 3: I find the text in the legends on the bottom of the figure a little confusing and it took me a while to decipher it just by looking at the figure:

I would suggest expanding the legend belonging to panels I-L a bit more, perhaps with an explicit/longer text such as “Prob. of QTP-binary (assuming qpAdm p-value ≥ 0.05 & admixture proportions [0, 1])”. Or, if this would be deemed to long, something slightly more explicit because the current “(QTP - binary) $p \geq 0.05$ & weights [0:1]” is too terse.

I also suggest replacing “N: FP qpAdm models” with more conventional “# N of qpAdm models” in the equation using the convention of # to represent counts (if this is indeed what is intended by “N:”).”

We have made the suggested amendments to the Figure 3 legend (see below).

Figure 3

Simple demographic models and qpAdm test performance (QTP). (A-D) Topological structures of the four simple demographic models. (E-H) QTP and number of plausible qpAdm models across the range of median pairwise FST values calculated on the S1, S2, R1, R2, and R3 populations. For each simulation iteration we represent the counts of the number of plausible single and two-source qpAdm models (21 is the maximum possible) with orange dots and the locally estimated scatterplot smoothing (loess) computed in R and shown with the orange line. We show the QTP value for each simulation iteration with blue dots and the loess smoothing with the blue line. (I-L) Logistic GAM probability for the QTP-binary response variable with admixture date (Tadmix) and median pairwise FST as predictor

variables. The gray dots are unique combinations of simulation parameters placed in the space of predictor variables. Vertical dotted lines in plots E-L show the median pairwise FST values at the approximate Iron (0.008), and Bronze Age (0.013) periods.

“line 178: Typo in the specified Ne value (10,00).”

Thank you, we have rectified this.

“line 424: When the authors write "subtle and weakly significant", do they talk about effect size? Otherwise, at a given significance level, something either is or isn't significant; talking about "weak significance" seems vague. I wasn't sure how to interpret this statement. Clarifying it with more explicit and rigorous language would help.”

We are thankful to the reviewer for identifying our unclear language. The text has been updated as shown below.

Lines 418-422:

However, in the presence of admixture to the source population from an outgroup, we observe a subtle overestimation of the S1 contribution to the Target (Model 2, delta-alpha mean = 0.01, one-sample T-test P-value = 0.02). However when admixture to S1 is from the iS2R2 branch, we observe an underestimation, albeit not significant, of almost equal magnitude (Model 3, delta-alpha mean = -0.01, one sample T-test P-value = 0.07) (Figure 5).

“line 464: The terms FPR and FDR appear to be first mentioned here but are not formally defined above in the context of admixture graphs. On line 481 a FPR definition is stated but it's not clear upon initial reading whether this definition is applicable to FPR discussed in the previous two paragraphs above this line.

Also, for formal reasons, the first instances of FPR and FDR on line 464 should be spelled out in their full non-abbreviated forms regardless of how obvious and generally known they are.”

We appreciate the reviewer's attention to detail. The definitions of the False Discovery Rate (FDR) and False Positive Rate (FPR) in the section titled "Complex admixture history of sources and qpAdm demographic inference" were previously provided (and remain unchanged) on lines 316-324 as follows:

We do observe subtle differences in their average performance for metrics such as False Positive Rate (FPR) = $FP / (FP+TN)$, False Discovery Rate (FDR) = $FP / (FP+TP)$, QTP, and QTP-binary (Table 1). From each simulation iteration, we computed the qpAdm FPR for each demographic Model as follows: we counted the number of plausible false qpAdm models (false positives: FP) to obtain the FP qpAdm model count. To obtain the number of true negative (TN) qpAdm models, we counted the number of rejected false qpAdm models. For example, in Model 1 simulation iteration 1,998, we have an FPR of 0.8 that occurred because, of the 21 total single and two-source qpAdm models, we have 16 FP qpAdm models and four false qpAdm models were rejected ($FP / (FP+TN) = 16 / 20$). We computed the FDR in the same fashion.

When we subsequently discuss the FPR and FDR within the section titled "qpAdm plausibility criteria and improving model inference accuracy," we have provided the following text to enhance clarity:

Lines 481-483:

We observed a substantial decrease in the average error rates (FPR and FDR, as defined above), and an increase in average performance metrics (QTP, and QTP-binary) across all demographic Models when introducing the admixture weight [0,1] constraint to the plausibility criteria in qpAdm (Table 1).

Lines 492-503:

We also evaluated the impact of requiring all single-source qpAdm models to be rejected on the FPR and FDR error

rates. Under this condition, we updated our FPR calculation as follows: For each simulation iteration, if at least one false single-source qpAdm model was plausible, all two-source qpAdm models were rejected and we then computed the FPR as the FP/(FP + TN) following the guide above. Meaning, a two-source qpAdm model can only contribute to the FPR if all single-source qpAdm models are rejected in its simulation iteration. Recalling the above example, in the demographic Model 1 simulation iteration 1,998, we had an FPR of 0.8 that occurred because, of the 21 total single and two-source qpAdm models, we have 16 FP models, six of which are single-source models. However, because we now condition on all single-source models to be rejected, we have six false positives (single-source models) and 14 true negatives (rejected two-source). As such, conditioning on the rejection of all single-source qpAdm models in this instance, results in a reduction in the FPR from 0.8 to 0.3. The FDR was computed following the same procedure, where all two-source qpAdm models are rejected if a single-source model is plausible in their simulation iteration.

I was slightly surprised by seeing notation $[x:y]$ indicating that a number lies between x and y throughout the manuscript. Perhaps this is inspired by R vector syntax or something similar but why not use a more common notation for intervals in mathematics: $[x, y]$ for closed intervals and (x, y) for open intervals, and combinations of both for semi-closed intervals? On a related note, the F_{st} -bin columns in Figure 4 share right or left F_{st} boundaries, respectively (0.008 and 0.013) but it's not clear which boundary is included in which panel. Again, using semi-closed interval notation will make this unambiguous.

We appreciate the reviewer's spotting our incorrect use of mathematical notation. We have made the necessary corrections throughout the manuscript, and on the y-axis labels of Figure 2 (see above).

Literature cited

Harney, É., Patterson, N., Reich, D. & Wakeley, J. Assessing the performance of qpAdm: a statistical tool for studying population admixture. *Genetics*, Volume 217, Issue 4, April 2021.

Lawson, D.J., van Dorp, L. & Falush, D. A tutorial on how not to over-interpret STRUCTURE and ADMIXTURE bar plots. *Nat Commun* 9, 3258 (2018).

June 11, 2024

RE: GENETICS-2024-307001

Dr. Christian D Huber
The Pennsylvania State University
Department of Biology
State College, PA 16801

Dear Dr. Huber:

Congratulations! We are delighted to inform you that your manuscript entitled "**Testing Times: Disentangling Admixture Histories in Recent and Complex Demographics using ancient DNA**" is acceptable for publication in GENETICS. Many thanks for submitting your research to the journal. We appreciate your careful and well-structured response to the reviewer's suggestions, and the addition of several supplementary figures, which help clarify your results. This will be a valuable contribution to inference of human demographic history.

To Proceed to Production:

1. Format your article according to GENETICS style, as discussed at <https://academic.oup.com/genetics/pages/general-instructions>, and upload your final files at <https://genetics.msubmit.net>.
2. Your manuscript will be published as-is (unedited-as submitted, reviewed, and accepted) at the GENETICS website as an Advanced Access article and deposited into PubMed shortly after receipt of source files and the completed license to publish. Please notify sourcefiles@thegsajournals.org if you do not wish to publish your article via Advanced Access.
3. We invite you to submit an original color figure related to your paper for consideration as cover art. Please email your submission to the editorial office or upload it with your final files. You can submit a small-sized image for evaluation, and if selected, the final image must be a TIFF file 2513px wide by 3263px high (8.375 by 10.875 inches; resolution of 600ppi). Please avoid graphs and small type.

If you have any questions or encounter any problems while uploading your accepted manuscript files, please email the editorial office at sourcefiles@thegsajournals.org.

Sincerely,

Sohini Ramachandran
Associate Editor
GENETICS

Approved by:
Nicholas Barton
Senior Editor
GENETICS

note: Please add jnls.author.support@oup.com and genetics.oup@kwglobal.com (or the domains @oup.com and @kwglobal.com) to your email program's "safe senders" list. You will be contacted by both at various points during the production process.

Review comments (if applicable):

Reviewer #1 (Comments for the Authors (Required)):

I thank the authors for carefully addressing my comments and implementing my suggestions. The resubmitted manuscript represents a significantly improved version of an already excellent initial study. I have no further questions or comments.